# Bisnorgammacerane traces predatory pressure and the persistent rise of algal ecosystems after Snowball Earth

Lennart M. van Maldegem [1,2,16], Pierre Sansjofre[3], Johan W.H. Weijers[4], Klaus Wolkenstein [5,6], Paul K. Strother [7], Lars Wörmer[2], Jens Hefter [8], Benjamin J. Nettersheim [1,2], Yosuke Hoshino[1,17], Stefan Schouten[9,10], Jaap S. Sinninghe Damsté[9,10], Nilamoni Nath [5,11], Christian Griesinger [5], Nikolay B. Kuznetsov[12,13,14], Marcel Elie[15], Marcus Elvert[2], Erik Tegelaar[4], Gerd Gleixner [1] & Christian Hallmann [1,2]

Eukaryotic algae rose to ecological relevance after the Neoproterozoic Snowball Earth glaciations, but the causes for this consequential evolutionary transition remain enigmatic. Cap carbonates were globally deposited directly after these glaciations, but they are usually organic barren or thermally overprinted. Here we show that uniquely-preserved cap dolostones of the Araras Group contain exceptional abundances of a newly identified biomarker: 25,28-bisnorgammacerane. Its secular occurrence, carbon isotope systematics and co-occurrence with other demethylated terpenoids suggest a mechanistic connection to extensive microbial degradation of ciliate-derived biomass in bacterially dominated ecosystems. Declining 25,28-bisnorgammacerane concentrations, and a parallel rise of steranes over hopanes, indicate the transition from a bacterial to eukaryotic dominated ecosystem after the Marinoan deglaciation. Nutrient levels already increased during the Cryogenian and were a prerequisite, but not the ultimate driver for the algal rise. Intense predatory pressure by bacterivorous protists may have irrevocably cleared self-sustaining cyanobacterial ecosystems, thereby creating the ecological opportunity that allowed for the persistent rise of eukaryotic algae to global importance.

[1] Max Planck Institute for Biogeochemistry, Hans-Knoell-Str. 10, 07745 Jena, Germany. [2] MARUM - Center for Marine Environmental Sciences, University of Bremen, Leobener Str. 8, 28359 Bremen, Germany. [3] Laboratoire Géosciences Océan, Université de Bretagne Occidentale, UMR 6538, Place Copernic, 29280 Plouzane, France. [4] Shell Global Solutions International B.V., Grasweg 31, 1031 HW Amsterdam, The Netherlands. [5] Max Planck Institute for Biophysical Chemistry, Am Fassberg 11, 37077 Göttingen, Germany. [6] Department of Geobiology, Geoscience Centre, University of Göttingen, Goldschmidt-Str. 3, 37077 Göttingen, Germany. [7] Department of Earth and Environmental Sciences, Boston College, Weston, MA 02493, USA. [8] Alfred Wegener Institute, Helmholtz Centre for Polar and Marine Research, Am Handelshaven 12, 27570 Bremerhaven, Germany. [9] Department of Marine Microbiology and Biogeochemistry, Royal Netherlands Institute for Sea Research (NIOZ) and Utrecht University, PO Box 591790 AB Den Burg, The Netherlands. [10] Department of Earth Sciences, Utrecht University, PO Box 80.0213508 TA Utrecht, The Netherlands. [11] Department of Chemistry, Gauhati University, Guwahati 781014 Assam, India. [12] Geological Institute, Russian Academy of Sciences, Pygevsky 7, Moscow 119017, Russia. [13] Gubkin Russian State University of Oil and Gas, Leninsky Pr. 65, 119991 Moscow, Russia. [14] Schmidt Institute of Physics of the Earth, Russian Academy of Sciences, Bolshaya Gruzinskaya str., 10-1, Moscow 123242, Russia. [15] Petroleum Development Oman (PDO), PO Box 81 Muscat 100, Sultanate of Oman. [16]Present address: Research School of Earth Sciences, The Australian National University, 142 Mills Road, Canberra, ACT 2601, Australia. [17]Present address: School of Biological Sciences, Georgia Institute of Technology, 310 Ferst Drive NW, Atlanta, GA 30322, USA. Correspondence and requests for materials should be addressed to L.M.v.M. (email: Lennart.vanmaldegem@anu.edu.au) or to C.H. (email: Challmann@bgc-jena.mpg.de)

The Late Neoproterozoic global glaciations[1,2] marked a pivotal point for the evolution of life on Earth. While Tonian (1.0–0.72 Ga) sediments are predominantly characterised by bacterial remains, along with traces of unicellular eukaryotes, more complex organisms emerged during the Ediacaran (0.64–0.54 Ga), eventually leading to the evolution of animals and other large organisms of the Ediacara Biota[3–6]. In particular, the relationship between the termination of the Cryogenian (0.72–0.64 Ga[7]) Snowball Earth events, and the evolution of complex multicellular life has fuelled abundant discussion as these glaciations have variably been considered either as bottlenecks or as catalysts for the evolution of organismic complexity[2–5,8–10]. Several independent geochemical proxies point to a significant rise in environmental oxygenation after the Marinoan deglaciation (~635 Ma)[11–13], potentially triggered by a substantial influx of nutrients during rapid melting. This would have stimulated photosynthetic primary productivity and, along with increasing carbon burial, may have stoichiometrically augmented the concentration of free molecular oxygen in the atmosphere and oceans[14]. The massive input of glaciogenic detritus and nutrients, in particular phosphorus[15], was suggested as a key trigger for the rise of archaeplastid eukaryotic algae as the dominant primary producers in global marine ecosystems[5] since under nutrient poor conditions, as reconstructed for the pre-Cryogenian, cyanobacteria will typically outcompete algae and sustain their dominance through a positive feedback loop (Supplementary Note 1).

While eukaryotic green algae appear to have dominated primary productivity throughout the Ediacaran[16], little is known about the nature and response of life in the direct aftermath of the Snowball Earth events, or how the deglaciation and postglacial biology may have impacted global biogeochemical cycles. This is because the majority of cap carbonates, which conformably drape almost all Marinoan glacial diamictites[1], contain only traces of organic matter and are mostly too thermally mature for the preservation of specific molecular biomarker information. Cap carbonates were likely deposited very rapidly under highly elevated $pCO_2$, significantly elevated seawater temperatures[17] and increased marine alkalinity[1]. Yet, no satisfying explanation exists for the genesis of the primary dolostones that makes up the base of almost all globally observed cap carbonates[18] since the precipitation of dolomite is kinetically inhibited at non-evaporitic low-temperature conditions due to strong hydration of $Mg^{2+}$ ions in solution[19–21]. While secondary dolomitisation can be excluded in these cases[22], evaluating a potential biological role in dolomite formation hinges on the hitherto missing molecular organic information from basal cap carbonates. The lack of good solutions to the dolomite problem, i.e. a significantly elevated proportion of dolostones deposited during the whole Precambrian, has now occupied geoscientists for more than a century[23] and the origin of many sedimentary dolostone deposits remains controversial[24].

To shed more light on these issues we focused on primary[22] dolostones of the Mirassol d'Oeste Formation (Araras Platform, Brazil; Supplementary Methods) and overlying Guia Formation limestones, as they currently represent the only known moderately mature ($T_{max} < 440\,°C$) Marinoan cap carbonate that preserves organic matter sufficiently well to allow for molecular organic geochemical characterisation (Supplementary Methods). The lower ~11 m of this cap carbonate are barren of organic matter, consistent with an oxidising depositional environment, as suggested by its pink colour and low enrichment of authigenic redox-sensitive elements[13]. However, a return to increasingly reducing conditions is indicated by the sudden precipitation and successive drawdown of the Pb, U and Mo reservoirs[13], an increase in sedimentary pyrite[13], the preservation of organic

matter at 11.2 m and a drop in $\delta^{15}N$ values that suggest a change from predominant denitrification to nitrogen fixation[25] (Supplementary Methods).

## Results and discussion

**Hydrocarbon signature of the Araras cap carbonate**. About one meter stratigraphically below the point where redox conditions became favourable for the wider preservation of organics, we find exceptionally high quantities (>1000 ng/g total organic carbon (TOC)) of a previously unidentified terpenoid (Fig. 1) that is present as the dominant molecular constituent in the lower portion of the Araras cap dolostone. The highly unusual signature gives way to a more common pattern of sedimentary hydrocarbons—consisting of n-alkanes, (demethylated) hopanes and steranes—just 1.5 m stratigraphically up section. These hydrocarbons represent the first comprehensive snapshot of post-Snowball Earth biology and ecology. Through multi-stage preparative gas chromatography (GC) and high-pressure liquid chromatography (HPLC) 21 μg of the unidentified target compound was isolated at ~99% purity (Fig. 2). In brief, 5 Å molecular sieving of the saturated hydrocarbon fraction was followed by size-exclusion HPLC and preparative GC (Fig. 2a). Compound-specific carbon isotope analysis via GC-isotope ratio mass spectrometry revealed a stable carbon isotope value of −27.7‰ (± 0.4‰), which closely resembles the $\delta^{13}C$ values of the bulk organic matter (−27.2 ± 0.7‰)[25]. The isolated compound was tentatively characterised as a pentacyclic saturated hydrocarbon via GC-mass spectrometry (GC-MS), while high-resolution-electron ionisation-MS confirmed a molecular formula of $C_{28}H_{48}$ (m/z 384.3744 M$^+$). Complete structural elucidation was achieved using one-dimensional (1D) and two-dimensional (2D) microcryoprobe nuclear magnetic resonance (NMR) spectroscopy (Fig. 2d; Supplementary Methods)[26]. Detailed analysis of the results of correlated spectroscopy (COSY) experiments, heteronuclear single quantum coherence (HSQC) experiments, and heteronuclear multiple-bond correlation (HMBCs) revealed a symmetrical pentacyclic triterpane structure consisting of only six-membered rings with six methyl groups. HMBC indicated that the methyl groups are located in positions 4 (C-23 and C-24), 8 (C-26), 14 (C-27) and 22 (C-29 and C-30), establishing the unambiguous identification of the isolated compound as 25,28-bisnorgammacerane (BNG); Fig. 2d; Supplementary Methods).

**Relationship between BNG and gammacerane**. To establish the paleo-environmental significance of BNG, we studied a collection of 271 rocks and oils, spanning the past ~800 Myr of Earth history (Fig. 3; Supplementary Methods). While BNG is observed in both open marine (11.6%) and restricted marine to lacustrine (42.1%) samples irrespective of lithology, we note that rocks with positive BNG detection were predominantly deposited in the (sub)tropical zone (paleolatitudes of ca. 30°N/S[28]; Supplementary Data 1) and tend to display lithological signs of elevated salinities. Oils commonly represent composite mixtures of a stratigraphically integrated interval. While oils with BNG typically also contain elevated levels of the structurally similar component gammacerane (γ; Fig. 3; the values of γ/[γ + C30 αβ-hopane] are 0.13 ± 0.01 in non-BNG oils vs. 0.21 ± 0.02 in BNG-containing oils; n = 105; Supplementary Data 1), rock samples tend to contain either gammacerane or BNG (Fig. 3; Supplementary Data 1). This observation tentatively hints towards a mechanistic or ecological connection between the two components. Gammacerane, the sedimentary remnant of the membrane lipid tetrahymanol, is a commonly used marker compound in oil and source rock studies[29] that has been empirically connected to

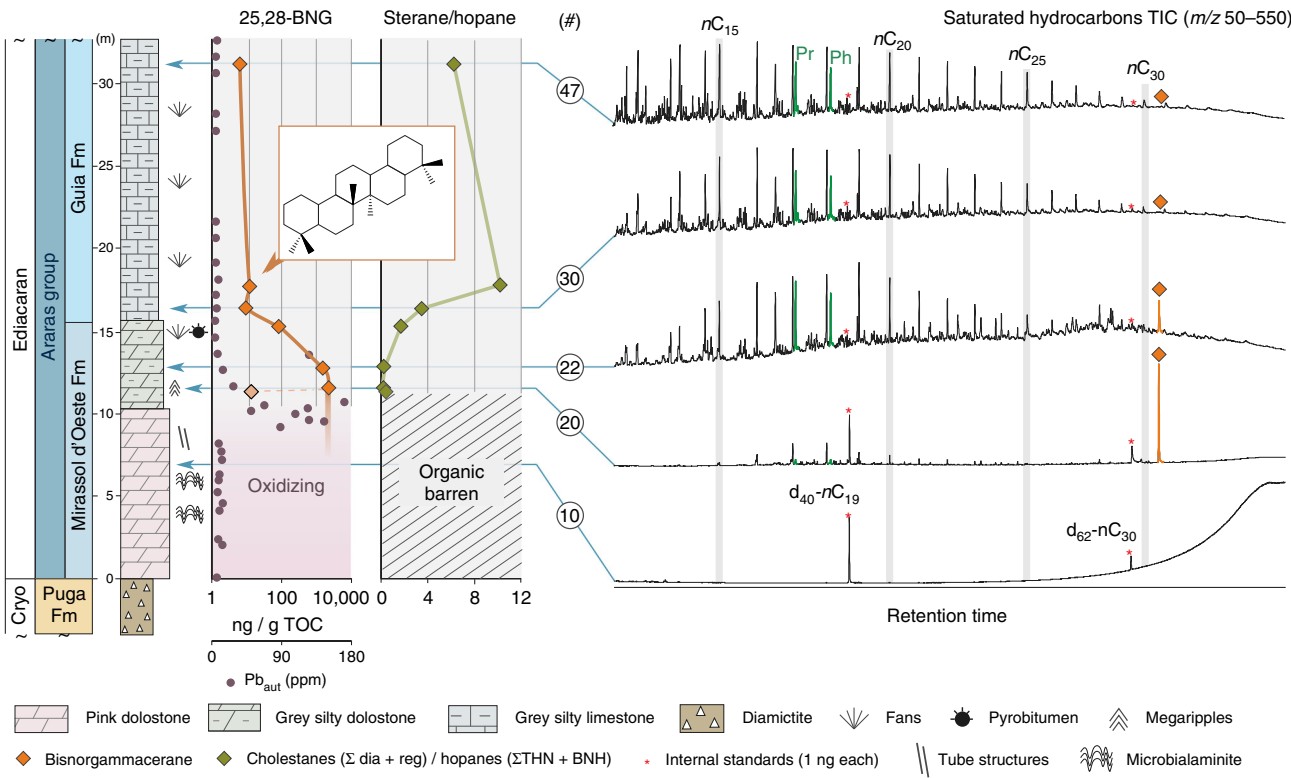

**Fig. 1** 25,28-bisnorgammacerane in the Araras cap dolostone. The lower Araras cap dolostones contain no preserved hydrocarbons (chromatogram #10), in agreement with an oxidising depositional environment. Following the precipitation and drawdown of redox-sensitive lead ($Pb_{authigenic}$) upon decreasing Eh, and allowing for organic preservation (around 11 m), the first observed dominant hydrocarbon biomarker (>1000 ng/g TOC, #20) after the Marinoan Snowball Earth has been identified as 25,28-bisnorgammacerane (BNG; orange peak). BNG decreases to values < 10 ng/g TOC (note the logarithmic scale) in the overlaying Guia Fm., coincident with the dolostone–limestone transition and parallelled by an increase of phototrophic (pristane and phytane peaks in green)-derived hydrocarbons and a strong increase of eukaryotic over bacterial biomarkers (cholestane ($\sum$dia + reg)/hopanes ($\sum$Ts + Tm + BNH); note that regular $17\alpha,21\beta$ (H)-hopanes were not detected in the Araras Group) both indicating an ecological change from a predominantly bacterial community to a phototrophic eukaryotic dominated ecosystem

strong water column stratification during deposition[27], or to conditions of elevated salinity[30]. Its biological precursor, tetra-hymanol, is predominantly biosynthesised by bacterivorous eukaryotic ciliates, as long as their diet is deprived of ster-ols[31,32]. From a phylogenetic perspective, these large hetero-trophic protists are among the earliest branching protozoans[33] and commonly thrive in low-oxygen environments near the chemocline[27,34,35]. An empirical link between stratification and gammacerane enrichment is predominantly based on Phaner-ozoic observations. Considering a lower availability of dietary sterols before the Ediacaran[5] and the wide variety of environments ciliates can inhabit, tetrahymanol biosynthesis by ciliates was likely less restricted by redox conditions prior to the global rise of eukaryotic algae.

The existence of a relationship between BNG and gammacer-ane is strengthened by observations of their distribution in rocks of the older, 0.74 Ga Chuar Group. Here average BNG abundances of $250 \pm 20$ ng/g TOC (30–520 ng/g; $n = 7$; Fig. 4) dwindle to zero in a narrow stratigraphic interval that records an increase of gammacerane from zero to an average value of $260 \pm 20$ ng/g TOC (45–870 ng/g; $n = 8$; Fig. 4). The same transitional interval exhibits a change in the stable carbon isotope offset between the $\delta^{13}C$ values of commonly fatty acid-derived $n$-alkanes (weighted average of $n$-$C_{15–33}$) and that of kerogen, i.e. amalgamated bulk organic matter ($\Delta\delta^{13}C_{A-K}$), with positive values (~7) dominating in the BNG interval and much lower values (~1) co-occurring with gammacerane (Fig. 4). Such drastic changes in $\Delta\delta^{13}C_{A-K}$ have been variably interpreted as

representing extensive heterotrophic reworking of primary produced organic matter[37], or as being reflective of a dominantly cyanobacterial *versus* eukaryotic primary producing community[36] (Supplementary Note 2).

When assessing the decrease of BNG in the context of changing $\Delta\delta^{13}C_{A-K}$ values (Supplementary Note 2) in the Chuar Group (Fig. 4) we can thus place its production into a bacterially dominated ecosystem, which is corroborated by an anticorrelative trend against increasing sterane abundances (Fig. 4). This information can be translated to the Araras cap carbonate, where we may tentatively reconstruct biological change from a bacterially dominated ecosystem in the direct aftermath of the Marinoan Snowball Earth glaciation to a mixed, or eukaryote-dominated ecosystem. This reconstruction does not only rely on BNG but is independently corroborated by an increase of eukaryote-derived steranes compared to bacterial hopanes in the Araras Group cap carbonate, from values < 1 to values > 5, which parallels the decline of BNG (Fig. 1). The observed absence of algae in the direct aftermath of the Marinoan glaciation is perhaps not surprising, since a large and turbid freshwater lens would have persisted in the world's oceans for tens of thousands of years[18] and ocean surface temperatures may have exceeded 40 °C[17]—both are conditions that are highly unfavourable for eukaryotic algal primary producers[10]. This could explain the cyanobacteria-dominated ecosystem that persisted during and immediately after the Marinoan deglaciation, as corroborated by a paucity of eukaryotic steranes, as well as BNG and its relationship to $\Delta\delta^{13}C_{A-K}$ dynamics.

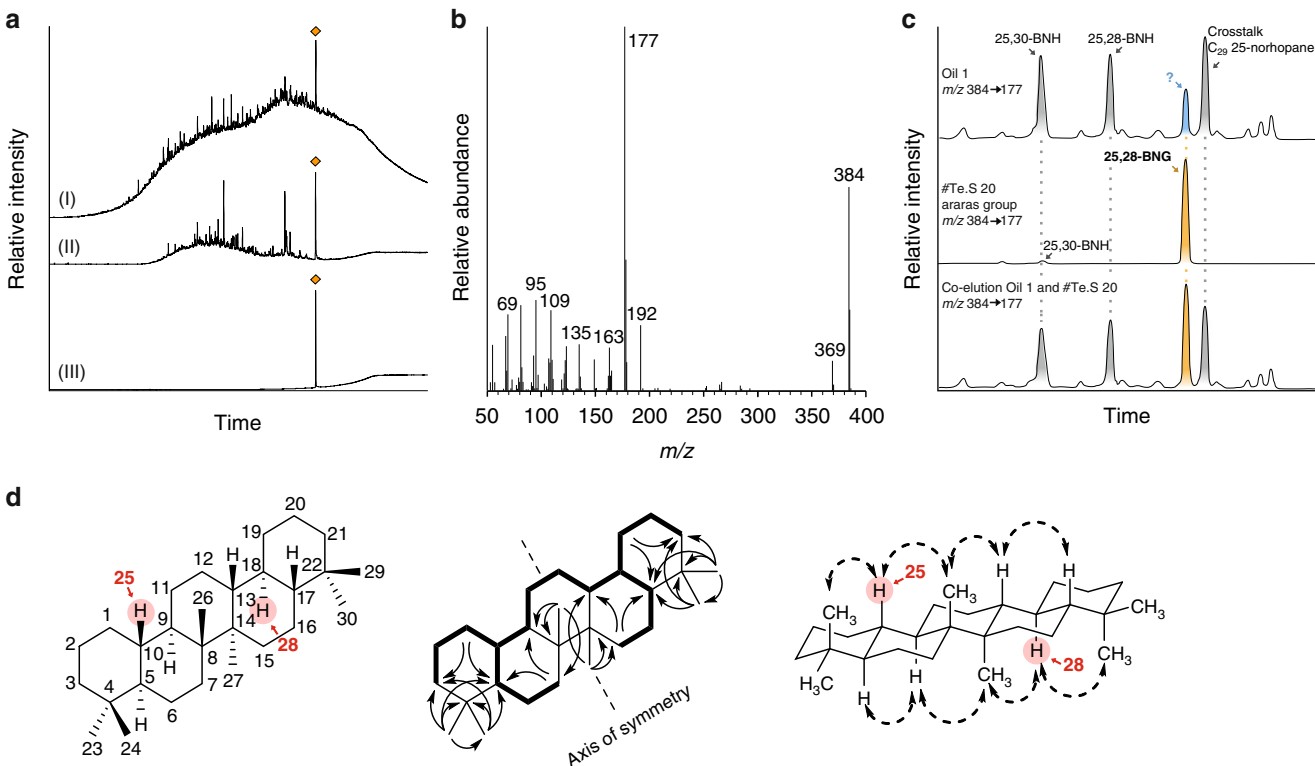

**Fig. 2** Isolation, characterisation and chemical structure of 25,28-bisnorgammacerane (BNG). **a** Total ion gas chromatography (GC)-mass spectrometry chromatograms (TIC, $m/z$ 50–550) showing the stepwise isolation of BNG (orange diamonds): (I) the initial saturated hydrocarbon fraction, (II) the remaining fraction containing BNG after molecular sieving and preparative liquid chromatography, and (III) the isolated fraction after preparative gas chromatography containing BNG at >99% purity. **b** Mass spectrum (electron impact: 70 eV) of BNG ($C_{28}H_{48}$, $m/z$ 384). **c** Co-elution experiment by tandem mass spectrometry ($m/z$ 384 → 177), using a GC-tandem mass spectrometry system equipped with a DB-1-MS column, of an oil sample (top) and a sample containing abundant BNG (Te.S 20; see Fig. 1 for identification), verifying the presence of BNG in the oil (bottom). **d** Chemical structure of BNG (5α(H),8β,9α(H),10β(H),13β(H),14α,17β(H),18α(H)); red numbers indicate positions 25 and 28 determined based on $^1$H-$^1$H correlated spectroscopy (bold), heteronuclear multiple-bond correlation (H → C) and nuclear Overhauser effect spectroscopy (also see Supplementary Methods)

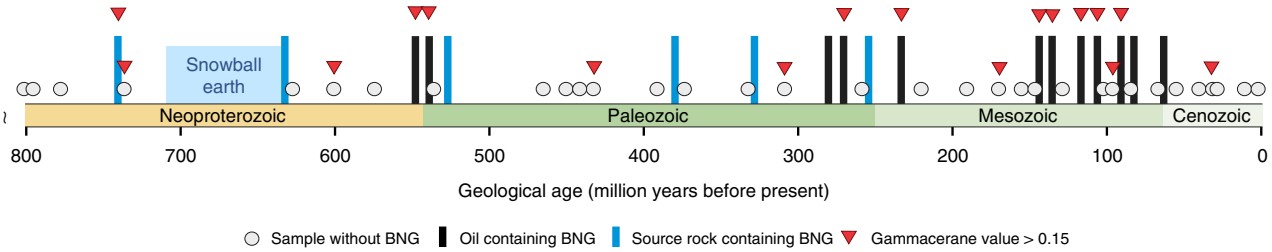

**Fig. 3** Secular occurrence of bisnorgammacerane and gammacerane. The molecular marker 25,28-bisnorgammacerane (BNG) occurs sporadically throughout the past 800 Myr of Earth history: positive identification of BNG in rocks indicated by blue bars, in oils shown by black bars; grey circles indicate absence of BNG in analysed samples. Red triangles indicate those samples with elevated gammacerane/($C_{30}$αβ hopane + gammacerane) values (>0.15; Supplementary Information), a molecular indicator for water column stratification[27], especially revealing elevated gammacerane values in many oils, which contain BNG (non-BNG oils 0.13 vs. BNG oils 0.21; $n = 105$; Supplementary Data 1)

**Origin of BNG: biosynthesis or degradation of gammacerane?**
The observed change in community composition in relationship to the diminishing BNG levels could be interpreted in a way that the BNG precursor is directly biosynthesised by bacteria or by non-sterol-producing eukaryotes, similar to tetrahymanol (the biological gammacerane precursor)-producing predatory ciliates that tend to thrive on bacteria. When envisaging a biosynthetic pathway analogous to that of tetrahymanol, the demethylated positions C-25 and C-28 in the BNG molecular structure (Fig. 2d) would correspond to the symmetric positions C-26 and C-29 in squalene (Fig. 5), the biosynthetic precursor molecule to tetra-hymanol. In contrast to the central methyl groups in squalene (i.e. C-27 bound to C-10 and C-28 bound to C-15) or the terminal

methyl groups (e.g. C-1 and C-25 bound to C-2), which all play crucial roles for proper folding during cyclisation to bacterial hopanes, the methyl groups corresponding to those missing in BNG have been shown to not affect hopanoid cyclisation[39,40]. This may allow us to speculate that BNG could indeed be cyclised via a bisnorsqualene intermediate, where demethylated tetrahymanol could yield a yet unknown physiological advantage. Hence, the direct biological formation of BNG cannot be categorically excluded and in this case 25,28-bisnortetrahymanol-synthesising organisms would have occupied a rare environmental niche where virtually no conventional terpenoids (i.e. sterols, hopanoids or tetrahymanol) were biosynthesised. Since BNG-producing organisms are not known and bisnortetrahymanol has not yet

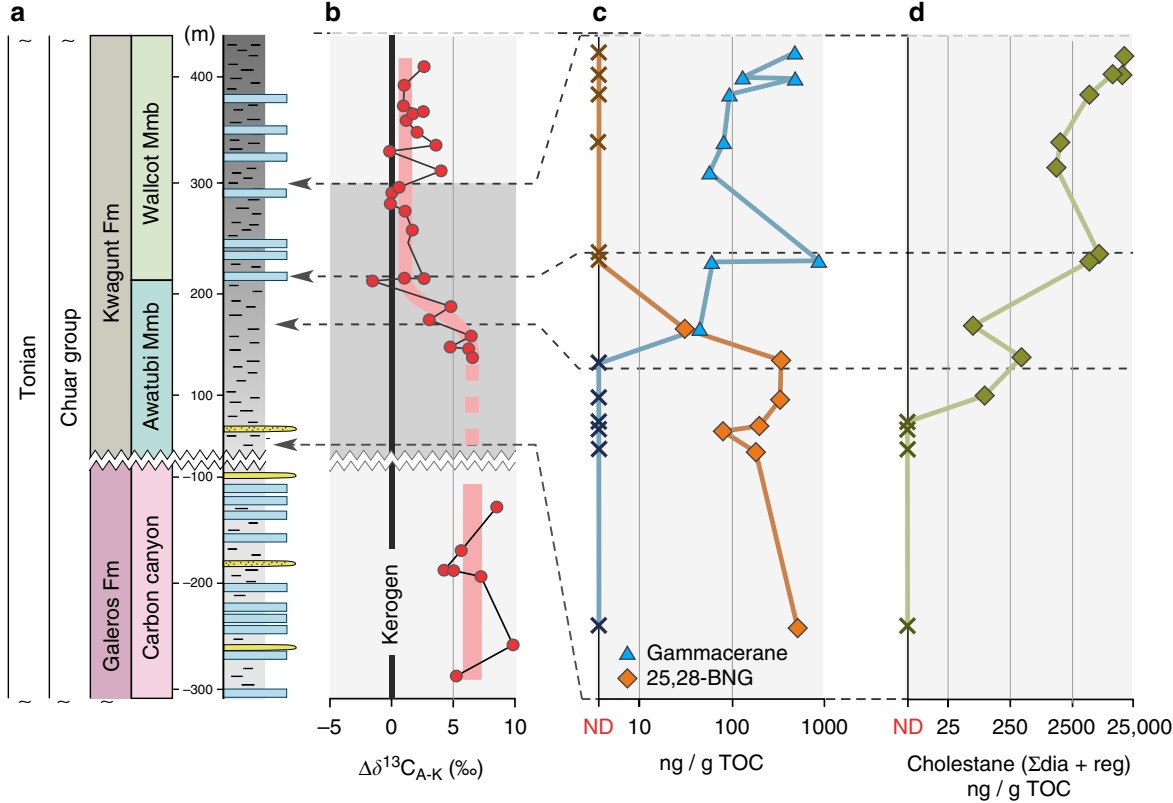

**Fig. 4** Heterotrophy, community change and 25,28-bisnorgammacerane. **a** Stratigraphy of the ~750 Ma Chuar Group; **b** the change observed in the $\delta^{13}C$ offset between kerogen and alkanes (noted as $\Delta\delta^{13}C_{A-K}$; here normalised to kerogen) parallels the **c** anticorrelative transition that is observed between gammacerane (blue triangles; ND not detected), a molecular fossil derived from tetrahymanol producing ciliates, and 25,28-bisnorgammacerane (25,28-BNG; orange diamonds); and **d** an increase in eukaryotic-derived cholestane (ng/g TOC; $\sum$dia + reg; green diamonds) indicating a shift in the composition of the primary producing community[36] and the intensity of heterotrophic reworking[37]

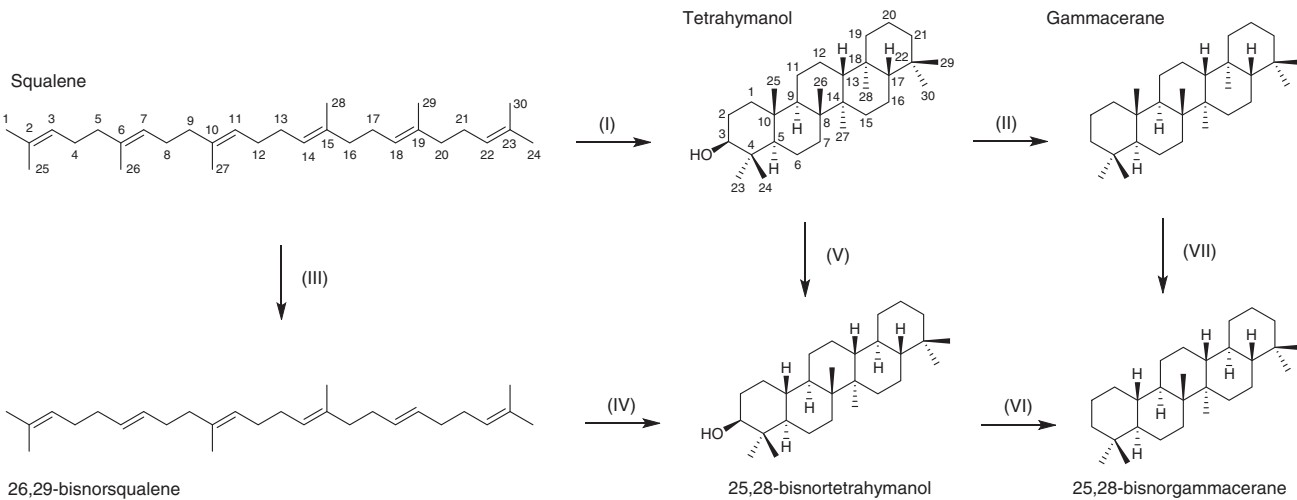

**Fig. 5** Possible formation pathways of 25,28-bisnorgammacerane (25,28-BNG). Squalene is cyclised to tetrahymanol via the tetrahymanol-synthases (THS) pathway[38] (I) present in eukaryotic ciliates[31], which after diagenesis (II) may become preserved as gammacerane (note the carbon numbering difference between squalene and triterpanes; e.g. C-26 and C-29 in squalene are identified as C-25 and C-28 in tetrahymanol). Hypothetically 25,28-BNG could form via the biosynthetic conversion of squalene to 26,29-bisnorsqualene (III), and a further cyclisation into 25,28-tetrahymanol (IV). However, thus far this is only a theoretical scenario, since the existence of the theorised biosynthetic pathways (III and IV) has not been described in the literature. Alternatively, BNG may stem from extensive microbial degradation of tetrahymanol prior to sedimentary burial (V), forming 25,28-bisnortetrahymanol, which could be preserved as 25,28-bisnorgammacerane after diagenesis (VI). Based on the preserved lipid signatures in the depositional environments we have no indications for a scenario where gammacerane is demethylated via post-depositional microbial degradation (VII) to form 25,28-BNG

been reported, post-Snowball seas could have witnessed alien ecosystems dominated by now extinct or marginalised organisms.

However, a direct biosynthetic source for BNG is purely hypothetical and would require the existence of yet undiscovered biosynthetic pathways. Additionally, the context of BNG occurrence causes interpretational problems. For example, one should ask why virtually no alkyl lipids—which are generally among the most abundant hydrocarbons in sedimentary organic matter—were preserved in samples that contain significant abundances of BNG (Fig. 1). The selective enrichment of polycyclic terpenoids in petroleum reservoirs is unambiguously attributed to biological degradation[41]. In analogy, the BNG anomaly may be explained by the selective enrichment of a particularly degradation-resistant molecule during microbial degradation in the water column or in sediments. This interpretation is supported by the co-occurrence of a range of demethylated hopane and gammacerane derivatives. Although the question remains, why biological degradation would almost exclusively affect the C-25 and C-28 alkyl substituents, since these bond properties do not drastically differ from those of other methyl substituents in tetrahymanol e.g. C-26 or C-27 (Fig. 5), cumulative evidence renders this explanation the most parsimonious. Organic matter preserved in the Araras Group contains a wide variety of terpenoids with similar demethylations to BNG, indicating that conditions were favourable for this type of degradation. Small, yet still significant, quantities of trisnorhopanes (Ts and Tm), 25,30-bisnorhopane (25,30-BNH, Fig. 2c) and an extended series of 25-norhopanes are present throughout the top ~25 m of sedimentary deposits in the Araras Group. The latter share a positional removal of the 25-methyl group with BNG, and also accompany BNG in the Chuar Group. Just like intact gammacerane is absent, also no regular $17\alpha,21\beta(H)$-hopanes $(C_{29-35})$ have been detected within this interval, pointing towards widespread demethylation of triterpenoid lipids. In addition, we observe tentatively identified 25-norgammacerane, as well as further dealkylated gammacerane derivatives in the Araras Group (Supplementary Figure 1), which highlights the significant variety in substrate demethylation. While all these nor-terpenoids may theoretically also represent direct biosynthetic products, the summed observation rather points towards post-biosynthetic dealkylation in the form of degradative biological processes. Elevated levels of pentacyclic triterpenoids demethylated at position C-10 (i.e. leading to 25-nor-terpenoids, including 25-norhopanes and BNG) are common in biodegraded oils and bitumens[29,41–43]. Their enhanced concentration may be due to the 25-methyl group being a preferred site for enzymatic attacks[44] in combination with the resulting compounds becoming concentrated over regular triterpanes because of enhanced resistance to further degradation[45]. Concentrations of both 25-norhopane as well as 25-norgammacerane were indeed found to increase during biodegradation of oils in the Liaoho Basin, where the degradation of gammacerane appeared to proceed slower than that of hopane[46].

As stated above, every investigated sample from the Araras Group that contains BNG also consistently features two unidentified triterpanes (Supplementary Figure 1). Their abundance was too low for NMR-based structural characterisation, but based on mass spectra, we tentatively identified these compounds as tetranorgammacerane ($M^+ = 356$; $C_{26}H_{44}$) and 25-nor-des-E-gammacerane ($M^+ = 316$; $C_{23}H_{40}$), where the latter could be a breakdown product of either gammacerane or hopanes. The loss of an E-ring in polycyclic triterpanes has been linked to photochemical[47] or microbial[48] alteration of the oxygenated ring of unsaturated lipid biomarkers[49,50], while tetranorgammacerane reveals the continuing progressive degradation of BNG. Both 25,30-bisnorhopane, which co-occurs with BNG in the Araras

Group, as well as 25,28-bisnorhopane, have been associated with reducing environments and the latter has been linked to microbes thriving at the chemocline in strongly stratified water bodies[51]. Notably, 28,30-bisnorhopanes appear to occur only as free compounds and are not present within the kerogen structure[52], which suggests that many of these demethylated compounds are not direct biosynthetic products, but that methyl groups are lost between production and sedimentary burial. The consistent co-occurrence of BNG with a large range of other demethylated terpenoids lends support to the idea that all these compounds have been generated through extensive bacterial degradation of biomass during deposition. This is analogous to the biological degradation of petroleum, where dealkylated polycyclic triterpenoids remain as the most stable molecular residues after the quantitative removal of alkanes and acyclic isoprenoids[53].

**Bacterially dominated ecology after the Marinoan glaciation.** If BNG is indeed formed through bacterial degradation processes, the removal of methyl groups must occur prior to diagenesis with a formation pathway likely leading from ciliate-biosynthesised tetrahymanol, via microbial degradation to 25,28-bisnortetrahymanol, and to 25,28-BNG after diagenesis (Fig. 5; Supplementary Note 2). This working model is again corroborated by the observed co-occurrence of BNG with positive $\Delta\delta^{13}C_{A-K}$ values in the Chuar Group and can be reconciled with both interpretations thereof[36,37]. Apart from reflecting the composition of the primary producing community[36], as discussed above, positive $\Delta\delta^{13}C_{A-K}$ have been previously attributed to the degree of heterotrophic reworking of primary produced organic matter[37], where bacteria degrade the phototrophic primary produced biomass, while adding their own, progressively more $^{13}C$-enriched[54] fatty acids to the sum of sinking organic matter (Supplementary Note 2). Although the results of a degradation model have indicated that multistage trophic cycling can only produce positive $\Delta\delta^{13}C_{A-K}$ in a narrow window of diagenetic conditions[36], this does not categorically exclude the possibility that positive $\Delta\delta^{13}C_{A-K}$ indicates a system where primary produced biomass undergoes strong heterotrophic reworking. In the context of the large variety of degraded terpenoids that co-occur with BNG, the observed synchronicity of the $\Delta\delta^{13}C_{A-K}$ change, from a large (~7‰) to a small (~1‰) offset, and the gammacerane–BNG transition in the Chuar Group (Fig. 4) indeed suggests that the extent of heterotrophic degradation of primary produced organic matter was an important component influencing both observations. In fact, this scenario also harmonises well with the alternative interpretation of $\Delta\delta^{13}C_{A-K}$, a change in its values reflecting a switch in the primary producing community from predominantly bacterial (positive $\Delta\delta^{13}C_{A-K}$) to dominantly eukaryotic (neutral $\Delta\delta^{13}C_{A-K}$), as supported by increased abundances of preserved sterane biomarkers in the Chuar Group (Fig. 4). If BNG is indeed derived from gammacerane, as suggested by its secular occurrence and the observed molecular trends in the Chuar Group (Figs. 3 and 4), the primary producing community thriving during deposition of the Araras cap carbonate must have initially been dominated by prokaryotes, which is corroborated by a paucity of steroids in the lower stratigraphic levels (Fig. 1). The intense dealkylation can only be plausibly explained with intense reworking of primary produced organic matter by heterotrophic bacteria, possibly as a consequence of slower sinking biomass, while ciliates would cease biosynthesizing gammacerane once they start acquiring sterols, derived from algal primary producers, through their diet.

**Protozoan predation and the Ediacaran rise of algae.** We suggest that sedimentary rocks of the post-Marinoan Araras Group

reflect an uncommon environment shaped by the intricate interplay of increasingly reducing conditions and a primary producing community that transitioned from being prokaryote-dominated to predominantly eukaryotic. Algae likely rose to ecological prominence[5] during the terminal Cryogenian, as indicated by the first co-occurrence of elevated sterane/hopane ratios and $C_{29}$ steranes in a cutting sample from a lime-mudstone horizon >635 Ma in age and likely situated within a Marinoan diamictite in Oman[55] (Supplementary Note 1). However, this single data point does not yet allow assessing the temporal and geographical extent of the algal ecosystems and may not yet reflect a persistent and global change in marine primary production. Strongly elevated surface water temperatures[17] in the direct glacial aftermath would have severely restricted eukaryotic primary producers[5], thereby allowing freshwater-derived[56] cyanobacteria to capitalise on elevated postglacial nutrient levels and to control primary productivity in the freshwater lens that formed as a consequence of massive glacial melting[17,18]. Despite the now elevated marine phosphorus levels, which have been considered a prerequisite for the rise of algae to ecological importance[5,15], our data confirm that cyanobacteria returned as the principal primary producers after the deglaciation. Under these conditions, cyanobacterial biomass-induced turbidity may have further contributed to the exclusion of eukaryotic algae due to their higher demand for light[10] (Supplementary Note 1). Cyanobacteria would have remained dominant, again maintaining a positive feedback loop[5] and preventing the establishment of algal-dominated ecosystems[10]. While Butterfield[10] suggested that the evolution of filter-feeding metazoa triggered the global ecological shift towards algal predominance by selectively removing the smaller cyanobacterial cells and their resulting turbidity, an alternative mechanism with similar outcome would have been the proliferation of bacterivorous grazers in the marine realm[5,57–59]. Just like in the modern ocean, more efficient protozoan bacteriophagy would have limited cyanobacterial population sizes, leading to higher levels of available surface water nutrients and creating a new, global ecosystem opportunity for eukaryotic algae[9]. We interpret the highly uncommon BNG signature in basal cap carbonates as reflecting the late stages of such massive bacterivorous grazing by ciliates that became visible once decreasing redox conditions, probably as a consequence of increased primary productivity, crossed the threshold of allowing for enhanced preservation of organic matter. These ciliates and their biomass were, in turn, largely degraded by heterotrophic bacteria in the deeper water column or sediments. The combination of intense feeding pressure by protists[5,57,58] and decreasing ocean surface temperatures[17] would have created the opportunity for eukaryotic algae to break through the self-sustaining feedback loop of cyanobacterial dominance[10], establishing global algal-dominated marine ecosystems.

The disappearance of BNG in parallel to rising sterane/hopane ratios, as well as its indirect connection to $\Delta\delta^{13}C_{A-K}$ reflects the rise of algae during the earliest Ediacaran, where increasing carbon export may have caused a drop in the intensity of heterotrophic reworking, and where tetrahymanol biosynthesis was muted by dietary sterol assimilation as a consequence of the increased availability of algal food sources. In the context of our data, we emphasise that the rise of algae did not occur as a direct consequence of enhanced nutrient availability. In the modern ocean, cyanobacteria and eukaryotic algae coexist even in the most oligotrophic regions, which is enabled by protozoan predation that limits the size of cyanobacterial populations[9,60,61]. While enhanced phosphorus availability may have been a prerequisite for the rise of algae, our data confirm that enhanced predatory pressure was likely needed to decimate the dominant cyanobacterial population that recurred after Snowball Earth[5,10]. Hence, bacterivorous feeding pressure by expanding

heterotrophic protists in the aftermath of the Cryogenian glaciations may have been the crucial trigger for the rise of algae. Thus, while the persistent global rise of eukaryotic primary producers probably only occurred during the earliest Ediacaran, the observed trends of BNG, gammacerane and cholestane in the Tonian Chuar Group can be interpreted as the oldest evidence for not just the presence, but an important role of predatory eukaryotes in modulating the primary producing community already prior to the Snowball Earth glaciations. The abundance of BNG in the Araras cap carbonates provides the first evidence for the pivotal ecosystem-engineering role of predatory protists in the direct aftermath of the glaciations.

**Biologically mediated dolomite in postglacial cap carbonates.** It has long been speculated that the environmental activity of certain heterotrophic bacteria plays a role in the precipitation of primary dolostones[22,62]. While these Mg-Ca carbonates can form in highly evaporitic environments or through secondary exchange with fluids[24], the formation of primary dolostones is kinetically inhibited under normal low-temperature marine conditions[20]. This raises an important question on the origin of the globally occurring postglacial cap dolostones, which were deposited in open marine settings and exhibit no signs of evaporitic conditions or of hydrothermal alteration. Recently, it was shown that in vitro microbial nucleation of Mg can overcome kinetic limitations—in particular, the large abundance of carboxyl groups[21] formed upon degradation of organic matter by bacterial heterotrophs (e.g. sulphate-reducing bacteria[63–65], methanogenic bacteria[21,66] and aerobic halophilic bacteria[67]) creates conditions conducive to the precipitation of dolostones by nucleating Mg-rich carbonates on their cell wall. Bacterial sulphate reduction at the sediment-water column interface in particular has been suggested to lead to the microbially mediated formation of primary dolomite[22]. While sulphur isotopes have indeed revealed a dramatic drawdown of marine sulphate levels in the aftermath of the Marinoan glaciation[68], this depletion and the departure from global steady-state conditions only started above the dolomite/limestone contact of the studied cap carbonate and are thus unlikely to have globally influenced the basal dolomite mineralogy. The observed abundances of BNG in the Araras cap dolostone that dwindle exactly at the dolostone–carbonate contact (Fig. 1) tentatively suggest a causal connectivity between the biological activity in the early postglacial ecosystem and the mineralogy of deposited carbonates. Although more work is required to establish direct mechanistic relationships, we propose that extensive heterotrophic utilisation of organic matter, both in the water column or sediment may have contributed to the dolomite lithology of basal cap carbonates. In this regard we hypothesise that not the degree of degradation may be the relevant factor, but extent of the process: the nearly complete (degradation-resistant compounds such as BNG appear to survive) respiration of a large mass of primary produced organic matter within a certain time period will generate significantly more molecular intermediates with acidic functionalities that can aid in stimulating the precipitation of dolomite, than the respiration of a smaller mass within the same time interval would do. While in both scenarios nearly all of the biomass will have been respired and the difference will be invisible from the sedimentary record, the mediation of other processes, such as e.g. dolomite precipitation could differ vastly. The dolomite problem may be a hallmark of pre-eukaryotic ecosystems, linked via carbon export, sinking speed and the susceptibility of primary produced organic matter to heterotrophic degradation.

In summary, a significant and consequential shift from dominantly bacterial to eukaryotic primary productivity during

the terminal Cryogenian has been linked to enhanced marine phosphate levels that rose as a consequence of glaciogenic detrital input[15]. Yet our knowledge of life in the immediate aftermath of the Snowball Earth glaciations is virtually nil. The well-preserved Araras cap carbonate provides the first snapshot of biology and ecology after the Marinoan glaciation. Low sterane/hopane values, and exceptional abundances of a novel and uncommon triterpenoid molecule that we identified as 25,28-BNG, suggest extensive heterotrophic reworking of biomass in a stratified, cyanobacterial-dominated ecosystem. In this regard BNG is a highly specific biomarker that probably derives from ciliates and reflects ecosystem dynamics, as supported by a systematic relationship to $\Delta\delta^{13}C_{A-K}$ dynamics in the older Chuar Group. In the Araras cap carbonate the demise of BNG coincides with a lithological change from dolostone to limestone, as well as with a relative increase of eukaryotic steranes. Extensive heterotrophic activity under specific environmental conditions may have contributed to the enigmatic global deposition of primary dolostones after the Cryogenian glaciations. More importantly, and in agreement with previous hypotheses[5,9,10,59], our data confirm that elevated nutrient levels were a prerequisite, but not the ultimate trigger for the ecological rise of eukaryotic marine phototrophs. We report a postglacial return to self-sustaining cyanobacteria-dominated ecosystems[10] that was likely favoured by elevated temperatures[5,17]. Patterns of BNG suggest that intense predatory feeding pressure by heterotrophic protists, may have sufficiently capped cyanobacterial population sizes—thereby decreasing turbidity and enhancing nutrient availability—to allow for the delayed eukaryotic invasion of this ecosystem space.

## Methods

**Workup of rock samples**. Prior to the laboratory workup all glassware, glass wool, silica gel, the stainless steel puck and mill, quartz sand and aluminium foil were baked at 500 °C for 8 h to remove any organic contaminants. Activated copper, used to remove elemental sulphur, was activated with a 1 M hydrochloric acid (HCl) solution before being washed with deionised (DI) water to reach neutrality and being cleaned three times with methanol (MeOH) and dichloromethane (DCM) under ultrasonication. High-purity-grade solvents (Merck n-hexane, cyclohexane, MeOH Suprasolv grade and DCM UniSolv grade) were used throughout all laboratory procedures.

In recent years it has become evident that hydrocarbon contamination from ancient and/or anthropogenic sources can pose a significant problem when analysing ancient rocks samples[69–75]. To overcome ambiguities and establish whether detected hydrocarbons are syngenetic and indigenous, solely indigenous (but not syngenetic) or derive from contamination introduced during residence, sampling or during the laboratory workup process, we separated all samples into interiors and exteriors, and analysed these in parallel to procedural blanks. Sufficiently large samples of a solid composition were sectioned using a lapidary trim saw (Lortone Inc.) fitted with an 8″ diamond-rimmed stainless steel saw blade that was previously cleaned by ultrasound-assisted solvent extraction (DCM) and by baking at 400 °C. Samples too small or too fissile for sectioning with a saw were abrasively separated into an exterior portion (in the form of abraded powder) and an interior portion using the micro-ablation technique[76]. Solid sample interiors and exteriors were wrapped in thick clean aluminium foil and fragmented into pieces smaller than ca. 1 cm³ by impact of a solvent-cleaned hammer. Powdering of these fragments was achieved in a Siebtechnik Shatterbox (Scheibenschwingmühle) using a custom-made stainless steel puck and mill, which were cleaned by baking (500 °C for 8 h) prior to use. Between samples, the puck and mill were cleaned by grinding and discarding clean Quartz sand (3×) followed by a solvent wash.

**Wet chemical sample processing**. Between 10 and 25 g of powdered sediment were extracted at 120 °C (20 min) under stirring with DCM in a CEM MARS 6 microwave extraction system. Three extractions were performed with 30 mL solvent each, which were pooled after centrifugation (2 min at 2000 rpm) in the Teflon extraction tubes. After removal of elemental sulphur using copper granules (activated with HCl$_{aq}$ [1 M], washed to neutrality [DI water] and solvent cleaned with MeOH and DCM), extracts were concentrated to a volume of 3 mL in a Büchi Syncore Analyst (700 mbar, 45 °C). Hexane (3 mL) was added and concentrated down again. This total lipid extract (TLE) was filtered through a glass wool-filled (ca. 0.5 cm) Pasteur pipette (baked clean at 500 °C) and allowed to evaporate to a volume of ca. 500 µL under ambient atmospheric conditions. One half of this TLE was removed for archival. The other half was transferred onto a silica gel (600 mg, 0.063–0.2 mm, Merck)-filled Pasteur pipette and fractionated into saturated

hydrocarbons, aromatic hydrocarbons and polars by sequential elution with n-hexane, n-hexane/DCM (7:3, v-v) and DCM/MeOH (1:1, v-v).

To obtain fractions of baseline-separated linear alkanes for the Chuar Group samples, alkyl lipids were isolated using a 5 Å molecular sieve (Merck). Adduction into 5 Å sieve (ca. 1 g) was performed overnight in cyclohexane (3 mL) at 80 °C. The non-adduct was extracted with cyclohexane (3 × 3 mL) under assistance of ultrasound. The adduct was released by digesting the sieve with HF$_{aq}$ (10 mL, 40%, Merck Millipore) under stirring for 2 h, followed by a liquid-liquid extraction with n-hexane (3 × 3 mL, Merck Suprasolv grade).

**Isolation of BNG**. BNG was isolated from 3.5 g of bitumen collected from the post-Marinoan Mirassol d'Oeste Formation, Araras Group, Brazil. After extraction, the saturated hydrocarbon fraction was cleaned up using a multi-step in-house protocol, where linear alkanes were removed by molecular sieving as described above, followed by liquid chromatographic preparation on an Agilent 1260 HPLC system to remove the unresolved complex mixture, and a final clean up by preparative GC. BNG was separated from interferences using size-exclusion chromatography (normal phase) at 22 °C eluting with DCM (1 mL/min) and isolated using an Agilent 1260 fraction collector. Preparative capillary GC was used to further purify the compound. Aliquots of 5 µL of the pre-cleaned hydrocarbon fraction in hexane were repeatedly injected via a Gerstel CIS 4 in solvent vent mode and into an Agilent 6890N gas chromatograph. The inlet was equipped with a deactivated, baffled glass liner (70 mm × 1.6 mm inner diameter (i.d.)), set to an initial temperature of 40 °C and heated to 320 °C at 12 °C/s and a final hold time of 2 min. For compound separation, the GC was equipped with a Restek Rxi-XLB capillary column (30 m, 0.53 mm i.d., 0.5 µm film thickness). The GC was operated using Helium as carrier gas at a constant flow of 4 mL/min. After an initial time of 2 min at 60 °C, the oven was heated with 20°/min to 150 °C and with 8 °C/min to 320 °C, with a final hold time of 6 min. A zero dead volume splitter diverted 1% of the column effluent via a restriction control capillary to a flame ionisation detector (FID), whereas the remaining 99% were transferred to a Gerstel preparative fraction collector (PFC). The PFC was connected via a heated fused silica transfer capillary to the GC and set to 320 °C. The switching device, directing the column effluent into seven time-programmable individual traps, was also set to 320 °C. After trapping, the compound was recovered with 5 × 500 µL hexane. External quantification using the response factor of squalane revealed a recovery of ~20.6 µg of the compound with a purity of ~99%.

**GC and MS**. Full scan analyses were performed on a Trace GC Ultra gas chromatograph (Thermo Scientific) coupled to an ALMSCO BenchTOF-dx mass spectrometer (MS). The GC was fitted with a VF-1 MS column (40 m, 0.15 mm i.d., 0.15 µm film thickness) using a constant flow (1.4 mL/min) of He (5.0, Westfalen AG) as a carrier gas. Samples (typically 1/1000 µL) were injected in splitless mode using a PTV injector (ramped from 60 to 315 °C at 14.5°/s). The GC oven was subsequently held at 60 °C (2 min) before ramping at 4.5°/min to a final temperature of 325 °C, which was held for 10 min. Ionisation was achieved at 70 eV (electron impact) and 250 °C with a filament current of ca. 4 A. Data were measured from m/z 30 to 800 but only recorded from m/z 50 to 550 at ca. 1000 mass resolution using 2469 scans per scanset and a scanset period of 250 ms. Analytes were quantified by comparison to internal standards perdeuterated $C_{30}D_{62}$ triacontane (Sigma-Aldrich) in the saturated hydrocarbon fraction and $d_{14}$-p-terphenyl (Sigma-Aldrich) for the aromatic fraction without correcting for individual response factors.

Target compounds analysis for biomarkers was performed on a Thermo Quantum XLS Ultra triple quadrupole MS coupled to a Thermo Trace GC Ultra, fitted with a DB-XLB capillary column (60 m, 0.25 mm i.d., 0.25 µm film thickness) and a deactivated pre-column (10 m, 0.53 mm i.d.). A constant flow of Helium (5.0, Westfalen AG) was used as a carrier gas (1.3 mL/min). Volumes of typically 1 or 2 out of 1000 µL were injected on column at 70 °C. The oven was held isothermal at 70 °C (5 min), then heated to 335 °C at 4°/min and held at final temperature for 9 min. Ionisation was achieved by electron impact at 70 eV and 250 °C, with an emission current of 50 µA. Q1 and Q3 were each operated in 0.7 Da resolution with a cycle time of 0.5 s. Q2 was operated with Argon 5.0 collision gas at a pressure of 1.1 mTorr and varying collision voltages depending on the target analyte. Compounds were quantified relative to $d_4$–5α-cholestane (Sigma-Aldrich) without correcting for individual response factors.

Exact mass determination of BNG was performed on a JEOL AccuTOF time-of-flight mass spectrometer equipped with an electron impact ionisation (70 eV) source. The sample was introduced via the direct exposure probe and perfluorokerosene was used for mass calibration.

The stable carbon isotopic composition of n-alkanes was determined at the Max Planck Institute for Biogeochemistry on a HP 5890 GC (Agilent Technologies) coupled to a Delta Plus XL isotope ratio mass spectrometer (Finnigan MAT, Bremen, Germany). Aliquots of 1 µL of the 5 Å adduct were injected in splitless mode at 280 °C and separated on a DB-1-MS capillary column (50 m, 0.32 mm i.d., 0.52 µm film thickness) operated with He (constant flow, 1.7 mL/min) as a carrier gas. The oven was held isothermal at 50 °C for 1 min, ramped at 9°/min to 308 °C, held isothermal for 2 min and finally ramped at 20°/min to 320 °C, where it was held isothermal for 3 min. Stable carbon isotope ratios were determined relative to a $CO_2$ reference gas, pulsed before and after the elution of analytes during a run.

This $CO_2$ was cross calibrated relative to a reference mixture of isotopically known n-alkanes. Data are presented in the conventional $\delta^{13}C$ notation as permil deviations from the VPDB standard. Samples were analysed in duplicate with analytical errors estimated at ±0.2‰.

The stable carbon isotopic composition of kerogens was determined at the Max Planck Institute for Biogeochemistry (IsoLab), Jena, Germany. Bulk powdered samples were digested with $HCl_{aq}$ (6 M, diluted from Alfa Aesar 36 % w/w aq.) to remove carbonates and washed to neutrality with DI water. After drying and homogenisation, the powdered samples (5–10 mg) were loaded into tin capsules (0.15 mL, 5 mm i.d., Lüdiswiss AG) and combusted online at 1020 °C in a Carlo Erba EA-1100 elemental analyser with a He carrier gas flow rate of 85 mL/min. Generated $CO_2$ was passed through a reduction furnace (650 °C) and separated from other gases on a Porapak PQ 3.5 GC column (80/100 mesh) at 40 °C. The product gases were transferred to a Finnigan MAT Delta + XL mass spectrometer via a ConFlo III interface operated in diluted mode for $\delta^{13}C_{org}$. The stable carbon isotopic values are reported in the permil notation relative to VPDB, after calibration by the NBS-22 reference standard.

**NMR spectroscopy**. Isolated BNG was dissolved in 40 μL 99.96 % $CDCl_3$ and transferred into a 1.7 mm NMR tube. 1D $^1H$ and 2D double quantum filtered COSY (DQF-COSY), nuclear Overhauser effect spectroscopy (NOESY), $^1H$-$^{13}C$ HSQC and $^1H$-$^{13}C$ HMBC spectra were recorded at 295 K on a Bruker Avance III 800 MHz spectrometer equipped with a 1.7 mm TCI CryoProbe. The $^1H$ and $^{13}C$ chemical shifts were referenced to $CHCl_3$ ($\delta_H = 7.26$ ppm, $\delta_C = 77.16$ ppm). NMR spectra were processed with Topspin 2.1 (Bruker).

DQF-COSY spectra were acquired as a $200^*(t_1) \times 4096^*(t_2)$ data matrices, where $N^*$ refers to N complex pairs, using 128 transients per FID and a 1 s delay between scans. Spectral width of 2000 Hz was chosen in $\omega_2$ and $\omega_1$, respectively. The time domain data were processed by zero filling to 8k and 8k points in the $\omega_2$ and $\omega_1$ dimensions, respectively, with a sine square window function in both dimensions.

HSQC spectra were acquired as a $512^*(t_1) \times 1024^*(t_2)$ data matrices, where $N^*$ refers to N complex pairs, using 176 transients per FID and a 1.5 s delay between scans. Spectral width of 2003 and 12195 Hz was chosen in $\omega_2$ and $\omega_1$, respectively. The time domain data were processed by zero filling to 2k and 1k points in the $\omega_2$ and $\omega_1$ dimensions, respectively, with a cosine square window function in both dimensions.

HMBC spectra were acquired as a $512^*(t_1) \times 1000^*(t_2)$ data matrices, where $N^*$ refers to N complex pairs, using 256 transients per FID and a 1.5 s delay between scans. Spectral width of 2000 and 12,066 Hz was chosen in $\omega_2$ and $\omega_1$, respectively. The time domain data were processed by zero filling to 8k and 2k points in the $\omega_2$ and $\omega_1$ dimensions, respectively, with a cosine square window function in both dimensions; 62.5 ms delay for evolution of long-range coupling, phase sensitive mode, no refocusing of long-range couplings before and no decoupling during acquisition.

NOESY spectra were acquired as a $256^*(t_1) \times 4096^*(t_2)$ data matrices, where $N^*$ refers to N complex pairs, using 40 transients per FID and a 2 s delay between scans. The mixing time used for the experiment was 400 ms. Spectral width of 2000 Hz was chosen in $\omega_2$ and $\omega_1$, respectively. The time domain data were processed by zero filling to 8k and 1k points in the $\omega_2$ and $\omega_1$ dimensions, respectively, with a cosine square window function in both dimensions.

**Global petroleum data set**. A sample set consisting of 94 petroleum fluids (Supplementary Methods) was studied at the Shell Laboratories in Rijswijk, the Netherlands. Saturate hydrocarbon fractions of crude oils were obtained by separation over $AgNO_3^-$-impregnated silica gel eluting with cyclohexane to obtain the saturate fraction and subsequently with toluene for the aromatic fraction. n-Alkanes were removed from the saturate fraction by adding activated (3 h at 120 °C) molecular sieve beads (5 Å, Merck) for at least overnight. The non-adduct, branched/cyclic (b/c) hydrocarbon fraction, was taken out and the beads rinsed three times with cyclohexane, which was added to the b/c fraction. These b/c fractions were analysed using GC/tandem mass spectrometry on an Agilent 7890A GC system coupled to an Agilent 7000 Triple Quadrupole Mass Spectrometer. Compound separation was achieved on a J&W DB-1 column (60 m × 0.25 mm i.d., 0.25 μm film thickness) using He as carrier gas and the following temperature programme: 50 °C (1 min)–4 °C/min–220 °C–1.2 °C/min–280 °C–3 °C/min–310 °C (20 min). The ion source was operated at 70 eV and 250 °C and the collision energy in the second quadrupole was set at 10 (arbitrary units). A selected set of parent-daughter mass transitions was scanned to monitor the compounds of interest. Compounds were identified by comparison of their retention times and elution order to standard samples. Presence of BNG in oil samples was confirmed by co-elution experiments of oils with sample TeS.20 (Araras Group, Brazil; Supplementary Methods).

## Data availability
Data repository: doi.pangaea.de/10.1594/PANGAEA.868689.

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

## Acknowledgements

We thank Arne Leider and Franziska Guenther for discussions; Paul Pringle and Xavier Prieto Mollar for laboratory support; Heike Geilmann, Holm Frauendorf, Györgyi Sommer-Udvarnoki, Aleksandra Poshibaeva and Mareike Neumann for analyses and data; the National Park Service (GRCA-00645) and Scottish National Wildlife Trust for sampling permissions; and Royal Dutch Shell for permission to publish. This work was funded by the Max-Planck-Society and the Deutsche Forschungsgemeinschaft (Research Center/Cluster of Excellence 309 - MARUM - Center for Marine Environmental Sciences; WO 1491/4–2 grant to K.W.; FOR 934 grant to C.G.) and by the "Laboratoire d'Excellence" LabexMER (Grant ANR-10-LABX-19). Sampling in Brazil was supported by CAPEX-COFECUB (442/04/06), sampling in Siberia and the Urals was supported by the Russian Ministry of Education and Science (14.Z50.31.0017 and 2017-220-06-304).

## Author contributions

C.H. and L.M.v.M. designed the research; L.M.v.M. conducted the organic geochemical workup, including preparative LC separation of the organic matter preserved in the carbonates rocks from the Araras Group and Chuar Group, and measured the organic extracts using GC-TOF-MS and GC-MS/MS; compound-specific stable carbon isotope analysis was conducted by L.M.v.M. and G.G.; J.W.H.W. and E.T. measured over a 100 additional Phanerozoic samples from the Royal Dutch Shell oil database. L.M.v.M., L.W. and M. Elvert performed preparative LC; J.H. operated the preparative GC. NMR spectroscopic measurements were performed by K.W. and N.N.; NMR spectra were interpreted by K.W., N.N. and C.G. to solve the structure and configuration of BNG; P.S., J.W.H.W., P.K.S., B.J.N., Y.H., S.S., J.S.S.D., N.B.K., M. Elie and E.T. provided additional samples, data and assisted with interpretation; L.M.v.M. and C.H. analysed all geochemical data and wrote the manuscript with input from all other authors.

## Additional information

**Competing interests:** The authors declare no competing interests.

