## [Peer Review File · Nature Communications]

Reviewers' comments:

Reviewer #1 (Remarks to the Author):

Van Maldegem and co-authors present an incredible and unprecedented biomarker record from an equally unusual sedimentary facies. The Marinoan cap carbonates are known for their puzzling array of most unusual sedimentary features such as megaripples, an isotopic excursion, crystal fans and tube structures that occur in the cap all around the world. The authors now show that the biomarker composition of a Marinoan cap carbonate is just as bizarre and puzzling as the rock they are found in. The base of the cap in the Araras Group is a pink oxidized dolostone devoid of hydrocarbons. However, at the transition to overlying grey silty dolostones a bitumen appears that consists nearly exclusively of a single compound that is new to science, bisnorgammacerane (BNG). Upsection, the signal transitions into a somewhat more familiar hydrocarbon assemblage – except that it is still very different from any other bitumen detected in the Ediacaran or any younger succession: it possesses a 100% predominance of the C27 sterane homolog cholestane.

The strangeness of these biomarker patterns, found in the arguable most unusual sedimentary rocks in the geological record, alone justifies publication in Nature Communication. But equally impressive is the length to which the authors went to isolate the new compound, BNG, and prove its structure beyond doubt using NRM technology – something that is not trivial with lean rocks such as these, and which has, to my knowledge, not been achieved before with hydrocarbons of this age. Also exemplary is the provision of a record of BNG through the entire Phanerozoic and into the Tonian to place the data into a broad temporal context.

The interpretation of the origin of BNG is well supported, particularly by the complementary Chuar Group record, and entirely plausible: a biomarker of heterotrophic ciliates that inhabited an intensely stratified water column in the post-Snowball ocean that was subsequently subject to intense microbial reworking.

More challenging is the interpretation of the unusual conditions that led to intense recycling of biomass in the post-Snowball ocean but preservation of BNG. Most sediments throughout the geological record contain virtually no organic matter and biomarkers, so complete reworking of biomass under most conditions is even more complete than in the Araras cap dolostone. Thus, intense and complete reworking of primary product as such is not really unusual. What is unusual here is intense reworking that leads to the removal of the precursors of almost all extractable organic compounds BUT preservation of a single degraded biomarker, BNG. The offered scenario does not fully explain why BNG is preserved and under what conditions that may have happened – but then, I can't think of any other plausible scenario. So, in a positive sense, the offered interpretation, and its connection with dolomite formation, will remain controversial and subject to intense debate – just as different models of cap carbonate formation and their many unusual features will remain controversial.

In summary, this is a paper that offers extremely unusual biomarker data, the most unusual biomarker data that I have seen, discovered in the most unusual sediments in the rock record. This, combined with a very thorough investigation of the structure of BNG, its origin and distribution through time, and a stimulating but controversial discussion about what it all means, makes this an exciting contribution. This paper should, without question, be published in Nature Communication.

Minor comments:

Line 56: References for statements in lines 56 to 60: I think the transition from a bacterial to a eukaryotic world in relation to Snowball glaciations is quantitatively best captured by the new observations in Brocks et al. (2018 in Nature) .

60: 'Ediacaran organisms' are any organisms living in the Ediacaran. I assume the authors refer to

the Ediacara Biota. The term 'true metazoans' is not defined – 'crown group metazoans'?

61: 'complex life' not defined. What exactly is meant here? the eukaryotic cell, complex multicellularity, animals?

102: probably should read 'suggests a mechanistic or ecological relationship'.

120: delete 'primary'. Ciliate OM is not primary.

122: Is 'pervasive' the right word here?

155: not fully correct: carbon burial controls atmospheric O₂ levels also on long time scales.

169: What does 'higher prevalence' of heterotrophic metabolism mean? High relative to what?

Relative to primary production (leading to less C burial in general), or relative to the Phanerozoic (i.e. more recycling within the microbial loop, but not necessarily leading to less total C burial?

Please specify. Also, the word prevalence is ambiguous – it has a spatial connotation, in the sense of 'heterotrophy is more widely distributed within ecosystems', which is not the intended use.

193: intense heterotrophy is not a rare conditions (see comment above: it rather is the norm – you need some different terminology to express what exactly is different and unusual in the cap carbonate)

200: a hydrocarbon cannot carry oxygen (because it would then not be a hydrocarbon)

SI

167: Figure 3, not 2.

Reviewer #3 (Remarks to the Author):

This manuscript is a bit of a Dr. Jekyll and Mr. Hyde story for me. On the positive side:

+ They have found a Sturtian glacial + cap carbonate sequence of low thermal maturity and good preservation. This is newsworthy by itself, as it opens the doors to a whole range of new studies that could tell us quite a lot about this very interesting period of Earth history.

+ They have found a new biomarker (BNG), and done amazing work to isolate it and prove its structure by NMR. This is an analytical tour de force, and – regardless of how the biomarker ends up being interpreted – this will forever be the paper that gets cited as introducing it. This part of the work kind of gets lost in the supplement, but to me it is the most solid and impressive part of the whole thing.

+ They have done admirable work searching proprietary sample databases for BNG in order to demonstrate a surprising temporal and latitudinal pattern of abundance,

+ the anticorrelation with gammacerane, the huge concentrations in the early strata of the cap carbonates, the correlation with changes in C isotopic composition of kerogen vs n-alkanes: these are all tantalizing evidence that this is indeed a biomarker with an important story to tell. All lines of evidence thus point towards this being the beginning of an important new biomarker tool for documenting environmental conditions.

On the more negative side, I just don't think the evidence for this representing "extreme heterotrophy" is that strong. It is plausible, yes, but certainly not conclusive or even beyond a reasonable doubt. Nor do I think the weight of the evidence is strong enough to then go ahead and extrapolate what extreme heterotrophy might mean for the global carbon cycle in the aftermath of a snowball glaciation. Here are some specific details:

- One line of evidence is to suggest that 25,28-bisnorgammacerane derives from gammacerane via the enzymatic demethylation of a parent gammacerane molecule by a heterotroph. One must then ask why degradation would stop there and leave the BNG product so abundant in the environment, and the proposed solution is that BNG is more stable. This is ... tenuous. It is hard to see from the structure why it would be so much more stable than everything else that it is the only peak left in the chromatogram. And why there would sometimes be both molecules left preserved, and sometimes all gammacerane and no BNG. And why almost none of the mono-demethylated

gammacerane? The authors use the anticorrelation of BNG and gammacerane in the Chuar group as evidence for the relationship, and it is suggestive, but then why don't we see any gammacerane in the Brazilian sequence as BNG starts to decline? There also seems to be a suggestion that the heterotrophic conversion of gammacerane to BNG only happens during extreme heterotrophy, as though, when the bacteria get especially hungry, they finally eat those last two methyl groups. This is on extremely shaky biochemical grounds, and in fact I doubt there is any known pathway in which an organism can conserve energy from demethylating a triterpenoid. All the demethylations I know of require energy input. If there is a diagenetic relationship, it seems more likely that it is an enzymatic accident rather than a survival strategy. In summary: the evidence for connection between gammacerane and BNG via heterotrophs is inconclusive, at best.

- An alternative hypothesis, and one that does not connect so easily to 'extreme heterotrophy', is that some unique organism biosynthesized 25,28-bisnortetrahymanol, i.e. that BNG is a unique biosynthetic product and not the diagenetic product of gammacerane. I can't see any strong evidence against this hypothesis, so it should at least be seriously considered. In some ways it is simpler than the diagenetic hypothesis.

- The whole concept of 'extreme heterotrophy' is too imprecise to be useful. What does it mean? A large fraction of primary production is remineralized (in which case, extreme heterotrophy would be all over the modern ocean gyres)? There is a large flux of heterotrophic remineralization, requiring both high production and high consumption? There are many trophic levels of heterotrophs? The Close and Pearson (2011) paper, cited in the manuscript, is a good model for a better way to talk about such concepts. They define two variables, degradation extent and trophic efficiency; the latter determines how many trophic levels can exist simultaneously without running out of carbon.

- The other piece of evidence used to connect BNG to heterotrophy is the temporal correlation between BNG levels and the difference in $\delta^{13}\text{C}$ values of kerogen and alkanes. The latter is interpreted here, and elsewhere (Logan et al papers) as representing the degree of heterotrophic recycling of organic matter. Even if correct, this is still only circumstantial evidence. But more importantly, the Close and Pearson paper shows that there is only a very limited parameter space that can generate such large $\Delta\delta^{13}\text{C}$ values under the heterotroph model. In fact, their preferred explanation is that large differences between kerogen and n-alkane $\delta^{13}\text{C}$ represent a change in ecology of the primary producers, from eukaryotic algae to cyanobacteria. That seems to have been lost on the current manuscript, where Close and Pearson are cited as supporting the link to heterotrophy. I might be tempted to turn this around and use the change in $\Delta\delta^{13}\text{C}$ as evidence for a shift in ecology, and that the producer of BNG is somehow associated with cyanobacteria-dominated primary production.

- The evidence used to argue for a relationship between gammacerane, BNG, and heterotrophy comes from just one rock section, and of a different age than the cap carbonate sequence. The same relationship is not seen in the Sturtian cap carbonate sequence. With $n=2$, it is hard to know which one is the outlier.

Given my hesitations about the solidity of the connection between BNG and heterotrophy, it will surprise no-one to learn that I also think it imprudent to extrapolate this scenario to the Precambrian carbon cycle. I agree fully that there is a tantalizing connection to dolomite that should be explored. The evidence suggests that microbes could well be involved, although I would say the evidence equally suits a role for changing ecology as much as it does increasing heterotrophy. Regardless, unless/until the authors can define more precisely what 'extreme heterotrophy' means, it is hard to make quantitative predictions about how it would affect carbonate precipitation. My next comment (if we were going there, which I do not recommend) would then be that they need a rudimentary box model to make these predictions quantitatively. We care about the ratio of DIC to alkalinity being generated, not just # moles of CO_2 ; and we need to know about the timescales of precipitation, CO_2 efflux from the oceans, continental weathering and return of alkalinity and cations to the ocean, etc. There are multiple feedback loops, with different time constants, and it is hard (and risky) to try to simply intuit what the net result will be.

Other more minor complaints:

- I agree that higher $\delta^{15}\text{N}$ values are, in the modern ocean, usually attributed to denitrification. That assumption might be a little shakier in the precambrian, where the N cycle perhaps did not look quite the same in an anoxic ocean. Assuming it is correct for now though, the authors are calling for vast amounts (10-100 atmosphere's worth) of organic carbon to be remineralized with nitrate as the electron acceptor. I am pretty sure there is not that much nitrate in the whole modern ocean, and I might imagine there was even less in an intermittently anoxic ocean. Some attention should thus be paid to whether there is enough nitrate to fuel all this proposed carbon remineralization.
- "the secular co-occurrence of BNG and gammacerane" By my count, they co-occur 10 out of 17 times, or 59% of the time. Are you certain that is statistically different than random?
- Line 109, saying a 25-norhopane is demethylated at C-10 is, I predict, going to be confusing to a non-biomarker expert. Why would carbon atom #25 be bonded to #10? You could refer to a numbered structure, but it might be easier to just leave off the reference to C-10.
- Line 115-116, this is the Logan model for how to generate an isotopic offset. Seems unfair to cite Close and Pearson without mentioning their alternative model that invokes cyanobacterial primary production
- line 156-157, sentence seems to imply that silicate weathering plays a role in regulating redox balance, which is not true. I think this is just a problem with wording.
- Lines 161-164. Confusing, because "less drawdown" cannot "further intensify" silicate weathering. At worst it can fail to reduce weathering as much. More generally, while I understand (I think, its not super clear) the logic for saying increased heterotrophy drives carbonate precipitation, I am missing the connection to why that carbonate is dolomite? You say that this model solves the dolomite problem, but have lost me along the way somewhere.

Supplement:

- line 77, means that very few samples contained BAQC's and all of them were discarded? Or that lots of samples contained BAQC's and only a few were discarded? Wording is ambiguous.
- line 164, this is one spot that struck me as being on very shaky biochemical grounds. Are you suggesting that ciliates, when they are very hungry, will eat the methyl groups off of their own kindred triterpenoids? And that they stop synthesizing their own tetrahymanol (because there is no gammacerane in the rocks)? Is there any evidence at all for such a process in other organisms, or other molecules? This sounds somewhat fanciful.
- line 279 and 291, these two equations are not exactly an apples-to-apples comparison. In the first (O_2 as the electron acceptor) you are consuming acetate, the conjugate base of a weak acid; in the second, you are consuming glucose, which is neutral. The differences in alkalinity are thus due both to electron acceptor as well as to the choice of organic substrate, which is maybe a little misleading.
- line 299-300, is this backwards? How can raising the pH, combined with accelerated precipitation of dolostone, lead to less CO_2 dissolving in the oceans? If it is not backward then I am clearly confused.
- line 312, I think "There is little doubt..." is a bit presumptuous here. I have seen several talks, as well as the paper by Bristow and Kennedy (2008, GSA Bull), questioning whether it is possible to sustain such a huge DOC reservoir in a physically overturning ocean with O_2 in the atmosphere. I agree the Rothman model does currently seem to be more popular. But it is not uncontested.
- line 317, simply telling us how much CO_2 is released is not enough to predict an impact on carbonate deposition. Also need to know the alkalinity budget, as its the CO_2/alk ratio that really matters for carbonate.
- Figure 1: what is Pb_{aut} , and why is it on this figure? Needs an explanation, either in the caption or the supplement.

Response to the points raised by Reviewer #1

Van Maldegem and co-authors present an incredible and unprecedented biomarker record from an equally unusual sedimentary facies. The Marinoan cap carbonates are known for their puzzling array of most unusual sedimentary features such as megaripples, an isotopic excursion, crystal fans and tube structures that occur in the cap all around the world. The authors now show that the biomarker composition of a Marinoan cap carbonate is just as bizarre and puzzling as the rock they are found in. The base of the cap in the Araras Group is a pink oxidized dolostone devoid of hydrocarbons. However, at the transition to overlying grey silty dolostones a bitumen appears that consists nearly exclusively of a single compound that is new to science, bisnorgammacerane (BNG). Upsection, the signal transitions into a somewhat more familiar hydrocarbon assemblage – except that it is still very different from any other bitumen detected in the Ediacaran or any younger succession: it possesses a 100% predominance of the C27 sterane homolog cholestane.

We thank the reviewer for this positive assessment.

The strangeness of these biomarker patterns, found in the arguable most unusual sedimentary rocks in the geological record, alone justifies publication in Nature Communication. But equally impressive is the length to which the authors went to isolate the new compound, BNG, and prove its structure beyond doubt using NRM technology – something that is not trivial with lean rocks such as these, and which has, to my knowledge, not been achieved before with hydrocarbons of this age. Also exemplary is the provision of a record of BNG through the entire Phanerozoic and into the Tonian to place the data into a broad temporal context.

The uncommon nature of BNG demanded a thorough analytical approach—including unambiguous identification and a search for other occurrences—before allowing us to make any statements about its implications.

The interpretation of the origin of BNG is well supported, particularly by the complementary Chuar Group record, and entirely plausible: a biomarker of heterotrophic ciliates that inhabited an intensely stratified water column in the post-Snowball ocean that was subsequently subject to intense microbial reworking.

More challenging is the interpretation of the unusual conditions that led to intense recycling of biomass in the post-Snowball ocean but preservation of BNG. Most sediments throughout the geological record contain virtually no organic matter and biomarkers, so complete reworking of biomass under most conditions is even more complete than in the Araras cap dolostone. Thus, intense and complete reworking of primary product as such is not really unusual. What is unusual here is intense reworking that leads to the removal of the precursors of almost all extractable organic compounds BUT preservation of a single degraded biomarker, BNG. The offered scenario does not fully explain why BNG is preserved and under what conditions that may have happened - but then, I can't think of any other plausible scenario. So, in a positive sense, the offered interpretation, and its connection with dolomite formation, will remain controversial and subject to intense debate – just as different models of cap carbonate formation and their many unusual features will remain controversial.

We agree that this aspect was poorly formulated. Of course, exceptionally intense heterotrophy is not uncommon, but rather the rule in many depositional systems. The difference is, as you point out, is that mostly absolutely no OM is preserved under such conditions, whereas BNG forms an exception. While we still think that BNG is indicative for intense reworking of primary produced organic matter (even though not as 'extreme' as encountered much more frequently in geological history), its value lies in allowing us to recognize the presence and activity of heterotrophic protists in the direct aftermath of the Snowball glaciations. Protistan grazing has been speculated to have played a key role in the clearing of cyanobacterial ecosystems. With BNG we actually have evidence for this process!

In summary, this is a paper that offers extremely unusual biomarker data, the most unusual biomarker data that I have seen, discovered in the most unusual sediments in the rock record. This, combined with a very thorough investigation of the structure of BNG, its origin and distribution through time, and a stimulating but controversial discussion about what it all means, makes this an exciting contribution. This paper should, without question, be published in Nature Communication.

We again thank you for this positive assessment.

Minor comments:

Line 56: References for statements in lines 56 to 60: I think the transition from a bacterial to a eukaryotic world in relation to Snowball glaciations is quantitatively best captured by the new observations in Brocks et al. (2018 in Nature).

We have added the reference accordingly.

60: 'Ediacaran organisms' are any organisms living in the Ediacaran. I assume the authors refer to the Ediacara Biota. The term 'true metazoans' is not defined – 'crown group metazoans'?

We agree and this was changed.

61: 'complex life' not defined. What exactly is meant here? the eukaryotic cell, complex multicellularity, animals?

We apologize for any confusion and have adjusted it in the text.

102: probably should read 'suggests a mechanistic or ecological relationship'.

Correct, this has been adjusted

120: delete 'primary'. Ciliate OM is not primary.

This has been removed in the changed version of the manuscript.

122: Is 'pervasive' the right word here?

This has been removed.

155: not fully correct: carbon burial controls atmospheric O₂ levels also on long time scales.

Thanks for pointing out this mistake. This whole paragraph has now been deleted in response to comments made by Reviewer-3.

169: What does 'higher prevalence' of heterotrophic metabolism mean? High relative to what? Relative to primary production (leading to less C burial in general), or relative to the Phanerozoic (i.e. more recycling within the microbial loop, but not necessarily leading to less total C burial? Please specify. Also, the word prevalence is ambiguous – it has a spacial connotation, in the sense of 'heterotrophy is more widely distributed within ecosystems', which is not the intended use.

In the new version of the manuscript this aspect is toned down and the particular sentence is not present anymore.

193: intense heterotrophy is not a rare conditions (see comment above: it rather is the norm – you need some different terminonology to express what exactly is different and unusual in the cap carbonate)

We agree and have decided to remove this statement.

200: a hydrocarbon cannot carry oxygen (because it would then not be a hydrocarbon)

This has been adjusted.

SI

167: Figure 3, not 2.

Thanks for noticing; this was changed.

Response to the points raised by Reviewer #3

This manuscript is a bit of a Dr. Jekyll and Mr. Hyde story for me. On the positive side:

+ They have found a Sturtian glacial + cap carbonate sequence of low thermal maturity and good preservation. This is newsworthy by itself, as it opens the doors to a whole range of new studies that could tell us quite a lot about this very interesting period of Earth history.

+ They have found a new biomarker (BNG), and done amazing work to isolate it and prove its structure by NMR. This is an analytical tour de force, and – regardless of how the biomarker ends up being interpreted – this will forever be the paper that gets cited as introducing it. This part of the work kind of gets lost in the supplement, but to me it is the most solid and impressive part of the whole thing.

+ They have done admirable work searching proprietary sample databases for BNG in order to demonstrate a surprising temporal and latitudinal pattern of abundance,
+ the anticorrelation with gammacerane, the huge concentrations in the early strata of the cap carbonates, the correlation with changes in C isotopic composition of kerogen vs n-alkanes: these are all tantalizing evidence that this is indeed a biomarker with an important story to tell. All lines of evidence thus point towards this being the beginning of an important new biomarker tool for documenting environmental conditions.

We thank the reviewer for recognizing the invested effort and the importance of our findings.

On the more negative side, I just don't think the evidence for this representing "extreme heterotrophy" is that strong. It is plausible, yes, but certainly not conclusive or even beyond a reasonable doubt. Nor do I think the weight of the evidence is strong enough to then go ahead and extrapolate what extreme heterotrophy might mean for the global carbon cycle in the aftermath of a snowball glaciation.

We partially agree with the reviewer and made significant changes to the manuscript in response to this point of critique. We still consider the majority of evidence pointing towards BNG being derived from gammacerane, this implying heterotrophic ciliates as a source organism. Furthermore, the presence of many other alkylated components as well as the near-absence of alkanes, suggests that we are witnessing a setting where a large portion of OM was degraded. This indeed has implications not only for the ecosystem, but more importantly for ecosystem change—a point that was also raised by Reviewer-3. We have completely removed the concept of 'extreme heterotrophy' (which, as also pointed out by Reviewer-1, is actually common throughout world's oceans). We have also removed the paragraph where we extrapolate this concept and speculate on the carbon cycle. In summary, we now provide an enhanced manuscript without speculative portions and with the focus shifted away from 'extreme heterotrophy' towards post-glacial ecosystem change.

Here are some specific details:

- One line of evidence is to suggest that 25,28-bisnorgammacerane derives from gammacerane via the enzymatic demethylation of a parent gammacerane molecule by a heterotroph. One must then ask why degradation would stop there and leave the BNG product so abundant in the environment, and the proposed solution is that BNG is more stable. This is ... tenuous. It is hard to see from the structure why it would be so much more stable than everything else that it is the only peak left in the chromatogram. And why there would sometimes be both molecules left preserved, and sometimes all gammacerane and no BNG. And why almost none of the mono-demethylated gammacerane?

These are all valid points. However there are questions we can ask in turn. For example, if BNG indeed had a biosynthetic source, what happened to all of other membrane components, such as fatty acids? Why don't we see preserved alkanes? While we have now added a section discussing the possibility of BNG being a direct biosynthetic product, we still consider the

degradation route a more plausible scenario for the formation of BNG. We may also ask the same question in degraded petroleum reservoirs: why do we sometimes see exceptionally abundant 25-norhopanes and little else? As a matter of fact, the lower Araras Group carries many similarities to biodegraded petroleum: all 'easily-degradable' components are gone, while we observe a significant variety of de-alkylated terpenoids. From comments by Reviewer-3 further below, we gather that our statement was interpreted in a way that the ciliates *themselves* demethylate gammacerane. This is of course not what we meant; rather we assume a high degree of general bacterial reworking of OM in the water column: primary produced OM in surface waters is nearly-quantitatively degraded by ciliates thriving at the chemocline. The ciliate OM in turn is nearly quantitatively degraded by bacteria living in deeper waters or in the sediment. As soon as environmental conditions become more reducing—possibly as a consequence of enhanced carbon export—preservation is enhanced.

The authors use the anticorrelation of BNG and gammacerane in the Chuar group as evidence for the relationship, and it is suggestive, but then why don't we see any gammacerane in the Brazilian sequence as BNG starts to decline?

We interpret this to be a consequence of enhanced eukaryotic primary productivity: when ciliates acquire dietary sterols they stop biosynthesizing tetrahymanol. Even though fluctuations in bacteria-vs-eukaryote primary producers existed prior to the Cryogenian (as e.g. indicated by $\Delta \delta^{13}\text{C}$ change in the Chuar Group), eukaryotes were quantitatively less significant prior to the Ediacaran (Brocks et al., 2017; Hoshino et al., 2017), which would have resulted in significantly lower abundances of sterols being provided as a diet for ciliates.

There also seems to be a suggestion that the heterotrophic conversion of gammacerane to BNG only happens during extreme heterotrophy, as though, when the bacteria get especially hungry, they finally eat those last two methyl groups. This is on extremely shaky biochemical grounds, and in fact I doubt there is any known pathway in which an organism can conserve energy from demethylating a triterpenoid. All the demethylations I know of require energy input. If there is a diagenetic relationship, it seems more likely that it is an enzymatic accident rather than a survival strategy. In summary: the evidence for connection between gammacerane and BNG via heterotrophs is inconclusive, at best.

We have removed the concept of 'extreme heterotrophy'. A variety of heterotrophic bacteria, e.g. sulfate reducers, can however indeed demethylate hydrocarbons and use this carbon directly as an electron donor (e.g. Aitken et al., 2013; Aitken and Larter, 2003; Rueter et al., 1994; Widdel et al., 2000; Wilkes et al., 2003). This e.g. occurs in petroleum reservoirs or after oils spills and probably also leads to demethylated hopanes. Hence we don't see a reason why bacteria should *not* demethylate tetrahymanol / gammacerane. I feel that we are possibly misreading the reviewer's comments and we would be grateful for the reviewer's advice.

- An alternative hypothesis, and one that does not connect so easily to 'extreme heterotrophy', is that some unique organism biosynthesized 25,28-bisnortetrahymanol, i.e. that BNG is a unique biosynthetic product and not the diagenetic product of gammacerane. I can't see any strong evidence against this hypothesis, so it should at least be seriously considered. In some ways it is simpler than the diagenetic hypothesis.

We have now added a paragraph discussing this possibility.

- The whole concept of 'extreme heterotrophy' is too imprecise to be useful. What does it mean? A large fraction of primary production is remineralized (in which case, extreme heterotrophy would be all over the modern ocean gyres)? There is a large flux of heterotrophic remineralization, requiring both high production and high consumption? There are many trophic levels of heterotrophs? The Close and Pearson (2011) paper, cited in the manuscript, is a good model for a better way to talk about such concepts. They define two variables, degradation extent

and trophic efficiency; the latter determines how many trophic levels can exist simultaneously without running out of carbon.

We agree and have completely removed the concept of 'extreme heterotrophy'. Many sections were accordingly rewritten. Indeed, extreme heterotrophy is ubiquitous, where primary produced organic matter is quantitatively remineralized. The concept we were initially pursuing is the additional remineralization of glacially-derived organic matter, which could fuel heterotrophy in a way that remineralization exceeds primary production, creating a net positive CO₂ flux. However we agree that this concept also is unsubstantiated with the existing dataset. While we do adhere to the idea of intense reworking of organic matter, the current rewritten version of the manuscript carries a principal focus on ecosystem change after the Snowball Earth events that may have been driven by predatory ciliates, which we still consider the most plausible precursor organisms to BNG.

- The other piece of evidence used to connect BNG to heterotrophy is the temporal correlation between BNG levels and the difference in $\delta^{13}\text{C}$ values of kerogen and alkanes. The latter is interpreted here, and elsewhere (Logan et al papers) as representing the degree of heterotrophic recycling of organic matter. Even if correct, this is still only circumstantial evidence. But more importantly, the Close and Pearson paper shows that there is only a very limited parameter space that can generate such large $\Delta\delta^{13}\text{C}$ values under the heterotroph model. In fact, their preferred explanation is that large differences between kerogen and n-alkane $\delta^{13}\text{C}$ represent a change in ecology of the primary producers, from eukaryotic algae to cyanobacteria. That seems to have been lost on the current manuscript, where Close and Pearson are cited as supporting the link to heterotrophy. I might be tempted to turn this around and use the change in $\Delta\delta^{13}\text{C}$ as evidence for a shift in ecology, and that the producer of BNG is somehow associated with cyanobacteria-dominated primary production.

Many thanks for pointing this out. We fully agree and apologize for mis-citing Close & Pearson. As a matter of fact we think that in the Araras Group the two interpretations are not mutually exclusive. We probably indeed observe a significant ecosystem change that we now discuss as an important observation. During the 'bacterially dominated' portion, we however also observe enhanced degradation and thus we believe that there may be an intricate interplay of ecosystems and redox conditions that automatically entail more or less heterotrophic reworking. We hope that the current discussion in the reworked manuscript conveys this accordingly and satisfactorily.

The evidence used to argue for a relationship between gammacerane, BNG, and heterotrophy comes from just one rock section, and of a different age than the cap carbonate sequence. The same relationship is not seen in the Sturtian cap carbonate sequence. With $n=2$, it is hard to know which one is the outlier.

This is partially correct. The situation is not that we see relationships in one setting and not in the other (i.e. the Araras Group), but that we cannot establish some of the proxies for the Araras Group samples due to the absence of sufficiently abundant alkanes in the crucial stratigraphic interval. As discussed above we attribute the absence of gammacerane in the Araras Group to a larger presence of eukaryotic algae. Due to the highly uncommon nature of BNG we presume that it is not a widely produced marker and rather points towards highly specific conditions, which is why we consider it fair to interpret the Araras BNG signature with additional clues that come from the Chuar Group. This is currently the best we can do and we have no doubt that over the coming years, our understanding of BNG will be improved through further studies.

Given my hesitations about the solidity of the connection between BNG and heterotrophy, it will surprise no-one to learn that I also think it imprudent to extrapolate this scenario to the Precambrian carbon cycle. I agree fully that there is a tantalizing connection to dolomite that should be explored. The evidence suggests that microbes could well be involved, although I would say the evidence equally suits a role for changing ecology as much as it does increasing

heterotrophy. Regardless, unless/until the authors can define more precisely what 'extreme heterotrophy' means, it is hard to make quantitative predictions about how it would affect carbonate precipitation.

My next comment (if we were going there, which I do not recommend) would then be that they need a rudimentary box model to make these predictions quantitatively. We care about the ratio of DIC to alkalinity being generated, not just # moles of CO₂; and we need to know about the timescales of precipitation, CO₂ efflux from the oceans, continental weathering and return of alkalinity and cations to the ocean, etc. There are multiple feedback loops, with different time constants, and it is hard (and risky) to try to simply intuit what the net result will be.

We again agree with the reviewer and are grateful that she/he pointed this out. We have removed all speculative extrapolation, as well as any implications for the carbon cycle. As suggested by Reviewer-3, the rewritten manuscript principally focuses on ecosystem change after the Snowball Earth glaciations.

Other more minor complaints:

- I agree that higher d¹⁵N values are, in the modern ocean, usually attributed to denitrification. That assumption might be a little shakier in the precambrian, where the N cycle perhaps did not look quite the same in an anoxic ocean. Assuming it is correct for now though, the authors are calling for vast amounts (10-100 atmosphere's worth) of organic carbon to be remineralized with nitrate as the electron acceptor. I am pretty sure there is not that much nitrate in the whole modern ocean, and I might imagine there was even less in an intermittently anoxic ocean. Some attention should thus be paid to whether there is enough nitrate to fuel all this proposed carbon remineralization.

Absolutely. While we still consider the elevated d¹⁵N values as indicative for denitrification given that the redox conditions during deposition of the Araras Group were moving towards anoxia, we have removed any discussion connecting this metabolism to carbon remineralization.

- "the secular co-occurrence of BNG and gammacerane" By my count, they co-occur 10 out of 17 times, or 59% of the time. Are you certain that is statistically different than random?

This has been reworded.

- Line 109, saying a 25-norhopane is demethylated at C-10 is, I predict, going to be confusing to a non-biomarker expert. Why would carbon atom #25 be bonded to #10? You could refer to a numbered structure, but it might be easier to just leave off the reference to C-10.

As requested by Reviewer-3, we now also discuss the possibility of a direct biosynthetic source for BNG. In this context we also talk about demethylated squalene (as a potential precursor to BNG) and need to be specific. However we tried to explain the demethylations better (see e.g. lines 188–201: C-27 bonded to C-10 etc.).

- Line 115-116, this is the Logan model for how to generate an isotopic offset. Seems unfair to cite Close and Pearson without mentioning their alternative model that invokes cyanobacterial primary production

As mentioned above, we agree and apologize for the confusing citation, which has now been corrected.

- line 156-157, sentence seems to imply that silicate weathering plays a role in regulating redox balance, which is not true. I think this is just a problem with wording.

This whole section has been removed in response to comments and suggestions made by Reviewer-3

- Lines 161-164. Confusing, because “less drawdown” cannot “further intensify” silicate weathering. At worst it can fail to reduce weathering as much. More generally, while I understand (I think, its not super clear) the logic for saying increased heterotrophy drives carbonate precipitation, I am missing the connection to why that carbonate is dolomite? You say that this model solves the dolomite problem, but have lost me along the way somewhere.

This whole section has also been removed in response to comments and suggestions made by Reviewer-3

Supplement:

-line 77, means that very few samples contained BAQC's and all of them were discarded? Or that lots of samples contained BAQC's and only a few were discarded? Wording is ambiguous.

We apologies for any confusion, we have made adjustment in the text to clarify this.

- line 164, this is one spot that struck me as being on very shaky biochemical grounds. Are you suggesting that ciliates, when they are very hungry, will eat the methyl groups off of their own kindred triterpenoids? And that they stop synthesizing their own tetrahymanol (because there is no gammacerane in the rocks)? Is there any evidence at all for such a process in other organisms, or other molecules? This sounds somewhat fanciful.

We absolutely agree. What we refer to is common bacterial heterotrophic degradation of all biomass, including that produced by ciliates. Why more BNG is produced than nor-gammacerane, we don't know. However the large variety of dealkylated components strongly points towards biological degradation of biomass.

-line 279 and 291, these two equations are not exactly an apples-to-apples comparison. In the first (O₂ as the electron acceptor) you are consuming acetate, the conjugate base of a weak acid; in the second, you are consuming glucose, which is neutral. The differences in alkalinity are thus due both to electron acceptor as well as to the choice of organic substrate, which is maybe a little misleading.

This has been removed.

-line 299-300, is this backwards? How can raising the pH, combined with accelerated precipitation of dolostone, lead to less CO₂ dissolving in the oceans? If it is not backward then I am clearly confused.

This has been removed.

- line 312, I think “There is little doubt...” is a bit presumptuous here. I have seen several talks, as well as the paper by Bristow and Kennedy (2008, GSA Bull), questioning whether it is possible to sustain such a huge DOC reservoir in a physically overturning ocean with O₂ in the atmosphere. I agree the Rothman model does currently seem to be more popular. But it is not uncontested.

We are aware of the controversy surrounding the Rothman model and believe that this is a question that will not be resolved easily. However with the modified version of the manuscript, and the removed aspects of ‘intense heterotrophy’ and its extrapolation, we also do not discuss the marine DOC reservoir anymore.

- line 317, simply telling us how much CO₂ is released is not enough to predict an impact on carbonate deposition. Also need to know the alkalinity budget, as its the CO₂/alk ratio that really matters for carbonate.

Correct. This has been removed.

- Figure 1: what is Pb_{aut}, and why is it on this figure? Needs an explanation, either in the caption or the supplement.

An explanation for the abundance of authigenic lead enrichment (decreasing redox conditions) has been added to the figure caption of figure 1, and additional references have been added to the statement in lines 107–110.

References

- Aitken, C. M. & LARTER, S. Anaerobic hydrocarbon biodegradation in deep subsurface oil reservoirs. *Nature* (2003).
- Aitken, C. M. *et al.* Evidence that crude oil alkane activation proceeds by different mechanisms under sulfate-reducing and methanogenic conditions. *Geochimica et Cosmochimica Acta* 109, 162–174 (2013).
- Brocks, J. J. *et al.* The rise of algae in Cryogenian oceans and the emergence of animals. *Nature* 548, 578–+ (2017)
- Hoshino, Y. *et al.* Cryogenian evolution of stigmasteroid biosynthesis. *Science Advances* 3, e1700887 (2017).
- Rueter, P. *et al.* Anaerobic oxidation of hydrocarbons in crude oil by new types of sulphate-reducing bacteria. *Nature* 372, 455–458 (1994).
- Widdel, F. & Rabus, R. Anaerobic biodegradation of saturated and aromatic hydrocarbons. *Current Opinion in Biotechnology* (2000).
- Wilkes, H. *et al.* Formation of n-alkane-and cycloalkane-derived organic acids during anaerobic growth of a denitrifying bacterium with crude oil. *Organic Geochemistry* 34, 1313–1323 (2003).

Reviewers' comments:

Reviewer #1 (Remarks to the Author):

In the first round of reviews, I strongly recommended publication of this work based on the incredibly interesting and unprecedented biomarker composition detected in one of the most important and enigmatic sedimentary successions, the Marinoan cap carbonate. I was also impressed by the most rigorous isolation and analysis of the new biomarker BNG. All this is still valid for the new strongly modified version of the paper. In the original submission, I and reviewer 3 criticised the just-so nature of the interpretation of the data, in particular the notion of 'extreme heterotrophy', the implications for dolomite precipitation and post Snowball climate. In the new version, the authors have largely replaced all of this with a discussion about the ecological implications of the findings. With this, the authors have eliminated most weak points of the previous version and turned the interpretation of the unusual biomarker record into a gripping, significant and in my opinion largely solid story (with some non-fatal problems noted below). I agree with the interpretation of BNG as a bacterially degraded ciliate marker, and the interpretation of the immediate post Marinoan ecosystem as dominated by bacterial primary production with abundant ciliates feeding on cyanobacteria. I also agree with the principle story about what happened next: cooling water and removal of bacteria by ciliates may have opened the ecosystem to eukaryotic primary producers (aka rise of algae). I regard this as an important story that should definitely and without question be published in Nature Communications.

Before the paper is published, I recommend addressing a couple of major and several minor flaws:

Most important point:

I do not think the authors really have evidence for a boost in algal abundances in the section that they studied (see more detail below – I think the transfer of the isotope discrepancy from the Chuar to the Araras without actual data from the Araras is not valid – particularly as there is no evidence for an algal boost in the Chuar either; this piece of evidence is theoretical and indirect and thus way too uncertain to draw a major conclusion). From Figure 1, I take that cholestane already occurs in the lowest sample in nearly maximal concentrations (10 ng/g) and there is not really an increase upsection. Moreover, the original work [ref 4] that places a rise of algal abundances into the Cryogenian also sees a jump in sterane diversity, and this is NOT seen in the Araras. At some point above the measured Araras section, sterane diversity will presumably increase to nearly modern levels, and I would consider such a change as the post-Ediacaran rise of eukaryotic phototrophs. However, I do not see this shift in the lower 30 m of the Araras analyzed here. This is not really a problem for the paper in my opinion, you just need to tone down the strengths of the conclusions and instead offer some honest speculation. This may be a plausible story line:

'ref [58a] speculated that ciliate proliferation after Snowball melting cleaned out bacterial phytoplankton, opening ecospace and freeing nutrients for algal radiation. Here we confirm this hypothesis and report massive abundances of ciliate biomarkers in the Marinoan cap carbonate'

This is an elegant confirmation of a previous hypothesis! You can then go on to state that algal proliferation is not seen in the studied section yet, but based on knowledge from other sections this must have occurred shortly after.

Some more comments in detail:

72-73: Here two conflicting concepts are mixed. Yes, under nutrient poor conditions, cyanobacteria outcompete algae. However, under such oligotrophic conditions, the water column will presumably not become turbid (nutrient poor waters will have low cell densities and are commonly clear) (please fix throughout the text unless you can cite evidence to the contrary)

75: Quote : "However, the post-Marinoan 'rise of algae' appears temporally decoupled from the

rise in marine phosphorus: a significant increase in the variability and overall mean P content of shales is already seen in the Cryogenian interglacial period and this suggests differences in phosphorus availability in the pre-Cryogenian versus the post-Tonian Earth system 11."

This statement puzzles me greatly (and it becomes only clear at the end of the paper what the authors claim here). Ref [4] places both the rise of algae and [P] increase into the Cryogenian. So I do not see the cited disconnect, and readers will not understand this either. More further below.

81: This should be 'Marinoan' Snowball Earth because biomarkers as well as fossils are known from sediments directly above the Sturtian diamictites. And this is also not quite correct for the Marinoan: there are e.g. the macroalgae from the diamictites of the Nantou formation that were probably emplaced during the melting stage.

82: "One major reason therefore...." The statement that follows has no logical connection to the previous sentence.

96 remove comma

114: again, this should be Marinoan Snowball. Maybe also write 'one of the first' to acknowledge fossil discoveries.

169: this should read 'ever since the late Cryogenian'. Algal steranes were found in shales beneath the cap carbonate in Oman, so however these are interpreted, they must be Cryogenian.

172 quote: ". Given the ecological role that eukaryotes played prior to the Ediacaran 4,37,38, tetrahymanol production was more common and thus sedimentary gammacerane is encountered more frequently and is not restricted to the pervasively anoxic settings prohibitive to eukaryotes during the Phanerozoic."

What is this statement based on? In Figure 3, gammacerane occurs in 2 of 5 samples (40%), and in the Phanerozoic in 13 of 41 samples (31%). I doubt that this is statistically significant, particularly given the extremely low sample size in the Tonian. I would remove this claim, it is not needed.

And based on what data is gammacerane in the pre-Ediacaran not restricted to anoxic environments?

211: paucity of eukaryotic steranes: be specific, paucity based on what? Abundance or diversity? What are the sterane/hopane ratios? Would it generally not make more sense to plot steranes/hopananes rather than absolute abundances? I do not see this surge in steranes that you describe at the dolomite – limestone boundary (see also statement in caption Figure 1 that makes this claim). It starts at 10 ng/g and it ends at the same concentration, and in between is a low and a high outlier. I can't see a trend that follows the inverse of BNG abundances.

290: what are decreasing redox conditions? You mean increasingly reducing conditions?

317 to 319: I find this hard to digest: you have evidence that the isotope offset is caused by intense reworking in the Chuar, but use it at the same time to argue for increasing algal productivity in the Araras. What do sterane/hopane ratios do in either formation?

318: of, not or

319: 'was replaced'

325: placing the rise of algae into Marinoan deglaciation: I could not find any evidence that the steranes discovered by Love et al (2009) come from shales within a diamictide. From what is known, the shales are underlying the diamictide, and it is speculation whether the shales

themselves are underlain by more diamictite. However, I agree that you can speculate that these steranes are from the Snowball melting stage (as long as it is clear that it is speculation). But then these steranes would still PREDATE the BNG discovery in this paper. So in any case, two statements must be true: the first rise of algae is late Cryogenian, not Ediacaran, and the post Marinoan sees a reprise of mostly cyanobacterial conditions (as correctly stated throughout the paper). Correcting point 1 will not have an impact on the story, so I am not sure why the authors need to place the first algal appearance into the post Marinoan melting stage.

In lines 322 etc., the authors write: "In contrast to previous studies 4,58, we here posit that the rise of algae occurred not as a direct consequence of enhanced nutrient availability. Similar to the requirement for bioavailable oxygen for respiration 59,60, we consider enhanced phosphorus availability as a prerequisite but not as the trigger for the rise of algae. Instead, our data indicate that enhanced feeding pressure was needed to decimate the dominant cyanobacterial population10."

In the above quotation, the authors claim that reference 58 states that the rise of algae is a direct consequence of increasing nutrients. However, reference [58] in fact states the opposite: "In the immediate aftermath of the melting glaciers, rising nutrient levels hypothetically resulted in a massive increase in cyanobacterial phytoplankton numbers until bacteriovorous grazers, such as ciliates, radiated into the pelagic realm, capping bacterial biomass [22]. As in the modern ocean, this bacteriovorous activity would have made nutrients available for larger phytoplankton [71] — and it is this effect that we may recognize as surging algal steranes in the Cryogenian."

This is in fact exactly what the authors claim as their new and contrasting explanation. The authors should thus rather first cite the original idea of [58] that a proliferation of ciliates (or other bacteriovorous grazers) may have been the trigger for algal proliferation and THEN use their new data to show that the prediction in [58] may in fact have been correct. (see also earlier comment on this)

The dolomite story

The new version is much less speculative and thus much better, but there is still one statement that appears implausible to me and that can simply be removed without damaging the paper: you may simply want to state that bacterial heterotrophy in some form associated with BNG formation, such as specific sulphate reducers, may be associated with dolomitization. 'Enhanced heterotrophy' is a weak explanation for reasons criticized by both reviewers in the first round of reviews: the heterotrophy in these sediments is by no means 'enhanced' relative to many organic lean limestones. Heterotrophy is just very unusual in these samples, not more complete in magnitude. Moreover, it is totally unclear in which way ciliates would or could have been involved in dolomitization. Do you know of any other occurrence of dolomite where BNG or gammacerane are elevated? Shouldn't most dolomites contain BNG then?

365: Here the authors state that the shift from bacteria to eukaryotes occurred between the Sturtian and Marinoan. I agree with this, but it contradicts the earlier claim that it occurred after melting of the Marinoan Snowball.

367: "eukaryotic steranes remain low during the Snowball Earth interglacial 4"

I think it would be wrong to cite [4] for this statement. [4] claims that steranes increased in the 'interglacial'. Moreover, the word 'interglacial' is reserved for short warm phases within glaciations, and this terminology is probably not appropriate for a 15 million year hot phase.

377-378 (and other places): While I agree with reviewer 3 that the Close and Pearson interpretation for the isotope offset needs to be cited and discussed, the Chuar data may rather speak in favour of the Logan model: in the Chuar, there does not appear to be any evidence for increased algal primary productivity in parallel with the isotope shift (or is there? If yes, then this data needs to be presented). If there is in fact no evidence for a shift from bacterial to eukaryotic

primary productivity in parallel with the isotope shift, then this speaks against the Close and Pearson model (in this particular instance). By contrast, you do have evidence for heterotrophic reworking (25-nor hopanes) in parallel with the isotope shift, and this is actually quite a beautiful confirmation of the Logan model (at least in this data set). I think you should make a call for one or the other model when discussing the isotope data because there is no evidence that both mechanisms are at play reinforcing each other (as claimed in the SI text) – and it is simply not terribly likely.

To summarize the uncertainties in the current interpretation:

(1) The Chuar section shows a correlation between BNG and isotopic offset. There is evidence that degradation is responsible for this offset (Logan model) but no evidence for an increase in algal productivity (Close model).

(2) For the Araras section, it was not possible to measure isotopes, but you speculate that there may be the same correlation. Based on such a hypothetical correlation, you speculate that (a) heterotrophic reworking caused the BNG abundance (ala Logan) AND (B) that the hypothetical isotope shift would also have been caused by an increase in algal productivity (Close model) without sterane data to back this up.

There is no increase in sterane abundance and diversity in the Araras, and there is no data for an isotopic offset. Thus, the evidence is clearly too weak and speculative to conclude that there was an increase in algal productivity relative to bacterial primary productivity in the analyzed Araras section.

378: 'Enhanced heterotrophic reworking' is back. I would really remove it, it is just not true. Just cite highly unusual heterotrophic conditions.

380: See above, this is not a new idea. Rather state that you confirm the hypothesis by ref [58].

382: the idea of a return of cyanobacteria through elevated temperatures was postulated by ref [4], not [15]

384: Here you acknowledge other papers for the feeding pressure idea. Good.

423: remove 'intense heterotrophic reworking' and maybe state 'suggested to indicate ciliate activity'

424 after the MARINOAN Snowball Earth

Figure 4: this data is really incredibly cool!

Supplementary Material

125: world dominated by cyanobacteria: please check whether ref 23 really makes this statement.

143: ref error

144: same as in main text.

149: sizes

225: delete 'be'

In summary, there are a number of important points that need to be addressed, but overall this is a superb, important and well written paper that is a major contribution to the field. Based on the beautifully executed biomarker identification and importance of the unusual biomarker distribution, it will become a classic.

Reviewer #3 (Remarks to the Author):

The authors have done a considerable amount of re-writing to address the concerns of this reviewer (and others). Most importantly, they have exorcised the problematic "extreme heterotrophy" references and included a discussion of the possibility of a direct precursor for BNG. I think they have struck a very nice balance in this regard. To be clear, there will probably still be significant debate in the community about their environmental interpretation of the BNG record. That is par for this course, and exactly what makes the manuscript so interesting. I think you should publish the manuscript as is. Its going to be a classic.

Reviewer #1 (Remarks to the Author):

In the first round of reviews, I strongly recommended publication of this work based on the incredibly interesting and unprecedented biomarker composition detected in one of the most important and enigmatic sedimentary successions, the Marinoan cap carbonate. I was also impressed by the most rigorous isolation and analysis of the new biomarker BNG. All this is still valid for the new strongly modified version of the paper. In the original submission, I and reviewer 3 criticised the just-so nature of the interpretation of the data, in particular the notion of 'extreme heterotrophy', the implications for dolomite precipitation and post Snowball climate. In the new version, the authors have largely replaced all of this with a discussion about the ecological implications of the findings. With this, the authors have eliminated most weak points of the previous version and turned the interpretation of the unusual biomarker record into a gripping, significant and in my opinion largely solid story (with some non-fatal problems noted below). I agree with the interpretation of BNG as a bacterially degraded ciliate marker, and the interpretation of the immediate post Marinoan ecosystem as dominated by bacterial primary production with abundant ciliates feeding on cyanobacteria. I also agree with the principle story about what happened next: cooling water and removal of bacteria by ciliates may have opened the ecosystem to eukaryotic primary producers (aka rise of algae). I regard this as an important story that should definitely and without question be published in Nature Communications.

We would like to thank the reviewer for this positive evaluation of our manuscript.

Before the paper is published, I recommend addressing a couple of major and several minor flaws:

Most important point:

I do not think the authors really have evidence for a boost in algal abundances in the section that they studied (see more detail below – I think the transfer of the isotope discrepancy from the Chuar to the Araras without actual data from the Araras is not valid – particularly as there is no evidence for an algal boost in the Chuar either; this piece of evidence is theoretical and indirect and thus way too uncertain to draw a major conclusion). From Figure 1, I take that cholestane already occurs in the lowest sample in nearly maximal concentrations (10 ng/g) and there is not really an increase upsection. Moreover, the original work [ref 4] that places a rise of algal abundances into the Cryogenian also sees a jump in sterane diversity, and this is NOT seen in the Araras. At some point above the measured Araras section, sterane diversity will presumably increase to nearly modern levels, and I would consider such a change as the post-Ediacaran rise of eukaryotic phototrophs. However, I do not see this shift in the lower 30 m of the Araras

analyzed here. This is not really a problem for the paper in my opinion, you just need to tone down the strengths of the conclusions and instead offer some honest speculation. This may be a plausible story line:

'ref [58a] speculated that ciliate proliferation after Snowball melting cleaned out bacterial phytoplankton, opening ecospace and freeing nutrients for algal radiation. Here we confirm this hypothesis and report massive abundances of ciliate biomarkers in the Marinoan cap carbonate'

This is an elegant confirmation of a previous hypothesis! You can then go on to state that algal proliferation is not seen in the studied section yet, but based on knowledge from other sections this must have occurred shortly after.

The key issue raised by the reviewer here is that we have no *direct* evidence for a rise in algal relevance. Indeed, cholestanes are already present in the lowest cap carbonate strata that also record BNG. Since the first submitted version of our manuscript was strongly geared towards extreme heterotrophy—a stray aspect that was highlighted by both reviewers during the previous review cycle—we did not focus on molecular indicators reflecting the ecological relevance of eukaryotes, such as the ratio of steranes over hopanes (S/H) as used by (Brocks et al., 2017). However, this data indeed exhibits a fascinating trend (Fig. 1): S/H start with values $\ll 1$ in the lowest strata and increase to values > 5 in parallel to the demise of BNG. Thus we indeed see a 'rise' of algae (i) in one and the same section and (ii) directly after the Marinoan deglaciation. Although we do not observe C29 steroids characteristic of Ediacaran green algae, photosynthetic red algae would have existed at least since ~ 1050 Ma (Butterfield, 2000; Gibson et al., 2017) and would be characterized by such a cholestane-dominated molecular signature. The question why the depositional environment of the Araras cap carbonate was dominated by red and not green algae is beyond the scope of this manuscript and is subject to ongoing studies.

To refrain from a too lengthy and detailed deviation of the key point, we can answer the main point of critique raised by Reviewer-1 with newly calculated data that has now been added to Figure 1: a systematic rise of the sterane/hopane ratio shows that phototrophic algae rose in ecological prominence throughout the rapid deposition of the cap carbonate.

Some more comments in detail:

72-73: Here two conflicting concepts are mixed. Yes, under nutrient poor conditions, cyanobacteria outcompete algae. However, under such oligotrophic conditions, the water

column will presumably not become turbid (nutrient poor waters will have low cell densities and are commonly clear) (please fix throughout the text unless you can cite evidence to the contrary)

Absolutely correct — thank you for pointing this out; we have adjusted this in the manuscript and moved the mention of cyanobacterial shading to the discussion of the post-Cryogenian ecosystem.

75: Quote : “However, the post-Marinoan ‘rise of algae’ appears temporally decoupled from the rise in marine phosphorus: a significant increase in the variability and overall mean P content of shales is already seen in the Cryogenian interglacial period and this suggests differences in phosphorus availability in the pre-Cryogenian versus the post-Tonian Earth system 11.”

This statement puzzles me greatly (and it becomes only clear at the end of the paper what the authors claim here). Ref [4] places both the rise of algae and [P] increase into the Cryogenian. So I do not see the cited disconnect, and readers will not understand this either. More further below.

We apologize for the confusion and thank you for pointing this out. We have (i) moved this aspect completely away from the introduction, where it would have confused readers, and (ii) amended it to minimize confusion: an initial rise of algae likely took place during the late Cryogenian, but ecosystems returned to cyanobacterial dominance after the Marinoan deglaciation. We posit that protistan predation was important in breaking this post-Marinoan dominance of cyanobacteria.

81: This should be ‘Marinoan’ Snowball Earth because biomarkers as well as fossils are known from sediments directly above the Sturtian diamictites. And this is also not quite correct for the Marinoan: there are e.g. the macroalgae from the diamictites of the Nantou formation that were probably emplaced during the melting stage.

Indeed. We have rephrased the sentence (Marinoan) and amended it. However we would like to point out that the mentioned samples from the Nantuo Fm represents shales within a Marinoan diamictite and hence reflect one component of terminal-Cryogenian biology. We believe that our detailed molecular dataset still presents the first ‘comprehensive snapshot of post-Marinoan biology and ecology’.

82: "One major reason therefore...." The statement that follows has no logical connection to the previous sentence.

Thank you for pointing this out, we have adjusted this sentence.

96 remove comma

Corrected

114: again, this should be Marinoan Snowball. Maybe also write 'one of the first' to acknowledge fossil discoveries.

This has been corrected in the new version.

169: this should read 'ever since the late Cryogenian'. Algal steranes were found in shales beneath the cap carbonate in Oman, so however these are interpreted, they must be Cryogenian.

We thank the reviewer for his comment, and we have adjusted it.

172 quote: ". Given the ecological role that eukaryotes played prior to the Ediacaran 4,37,38, tetrahymanol production was more common and thus sedimentary gammacerane is encountered more frequently and is not restricted to the pervasively anoxic settings prohibitive to eukaryotes during the Phanerozoic."

What is this statement based on? In Figure 3, gammacerane occurs in 2 of 5 samples (40%), and in the Phanerozoic in 13 of 41 samples (31%). I doubt that this is statistically significant, particularly given the extremely low sample size in the Tonian. I would remove this claim, it is not needed.

And based on what data is gammacerane in the pre-Ediacaran not restricted to anoxic environments?

We agree that this statement sounded too direct. We wanted to speculate on the possibility of having a higher probability of tetrahymanol production prior to the rise of algae. The sentence remains but has been strongly toned down to:

“Considering a lower availability of dietary sterols before the Ediacaran (Brocks et al., 2017) and the wide variety of environments ciliates can inhabit, tetrahymanol biosynthesis by ciliates was likely less restricted by redox conditions prior to the global rise of eukaryotic algae.” (lines 153–156)

211: paucity of eukaryotic steranes: be specific, paucity based on what? Abundance or diversity? What are the sterane/hopane ratios? Would it generally not make more sense to plot steranes/hopanes rather than absolute abundances? I do not see this surge in steranes that you describe at the dolomite – limestone boundary (see also statement in caption Figure 1 that makes this claim). It starts at 10 ng/g and it ends at the same concentration, and in between is a low and a high outlier. I can't see a trend that follows the inverse of BNG abundances.

We thank the reviewer for pointing this out; this was perhaps indeed not the best way of portraying the increase in eukaryotic derived biomarkers. We have re-created Figure 1, now plotting steranes *versus* the sum of present hopanes (i.e. desmethylhopanes), which display a fascinating trend, strengthening our overall argumentation: low values when BNG is abundant and a rise to elevated values when BNG significantly decreases.

290: what are decreasing redox conditions? You mean increasingly reducing conditions?

Indeed! We have adjusted this sentence.

317 to 319: I find this hard to digest: you have evidence that the isotope offset is caused by intense reworking in the Chuar, but use it at the same time to argue for increasing algal productivity in the Araras. What do sterane/hopane ratios do in either formation?

Apologies for causing any confusion, we have now also added sterane data for the Chuar Group to support our point that the top of the Awatubi member shows a change to a community with more eukaryotes. As we argue throughout the paper, we do not think that the two interpretations of Dd13C systematics (heterotrophy *versus* community structure) are mutually exclusive: due to slower sinking speed of bacterial cells, heterotrophic reworking of prokaryotic biomass will be higher than in a system where primary productivity is dominated by eukaryotes. Thus while Dd13C may be primarily controlled by community structure as suggested by (Close et al., 2011) this will likely

often indirectly entail enhanced heterotrophic reworking of primary produced OM when the latter is mostly bacterial and characterized by lower sinking speeds. In fact this is what we think can be seen in the Araras Group.

318: of, not or

Corrected

319: 'was replaced'

Corrected

325: placing the rise of algae into Marinoan deglaciation: I could not find any evidence that the steranes discovered by Love et al., 2009 come from shales within a diamictite. From what is known, the shales are underlying the diamictite, and it is speculation whether the shales themselves are underlain by more diamictite. However, I agree that you can speculate that these steranes are from the Snowball melting stage (as long as it is clear that it is speculation). But then these steranes would still PREDATE the BNG discovery in this paper. So in any case, two statements must be true: the first rise of algae is late Cryogenian, not Ediacaran, and the post Marinoan sees a reprise of mostly cyanobacterial conditions (as correctly stated throughout the paper). Correcting point 1 will not have an impact on the story, so I am not sure why the authors need to place the first algal appearance into the post Marinoan melting stage.

We agree with the reviewer. We have now reformulated the statement to reflect an initial rise of algae during the terminal Cryogenian (based on the data by Love et al. (2009)), followed by a return to a bacterially dominated ecosystem after the Marinoan deglaciation in Brazil; then under these specific environmental conditions, protistan predation may have allowed the renewed, now-persistent rise of algae to global dominance.

Nevertheless we would like to point out that the data by Love et al. (2009), interpreted by Brocks et al. (2017) to suggest that eukaryotic algal dominance was already established during the Cryogenian interglacial (elevated sterane/hopane and presence of C29 steranes) is based on only one (!) sample. One of two pre-Ediacaran samples from the SOSB was sandy/silty and likely contaminated, as pointed out by Love et al. (2009); they did not perform HyPy on this sample. The second sample was integer and contained C29 steranes in bitumen and in hydropyrolysates. However the stratigraphic position of this

sample is incredibly speculative: see Fig S1 in Love et al.(2009), where the lithology is not properly explained. In personal communication, Paul Hoffman pointed out that in the southern Mirbat area, the Cryogenian is expanded. Love's sample lies only ~200 m underneath the cap carbonate, whereas the whole Ghadir Manquil Fm. takes up about 800 m above the ca. 850 Ma crystalline basement. So while of course we cannot say anything confidently about the age of this sample, apart from that it is older than 635 Ma, the likelihood of it being pre-Marinoan (i.e. Cryogenian interglacial) is very slim in our opinion and this is an important aspect of SOSB stratigraphy and geochronology that deserves significant attention in the future. Independent of its age, calibrating the global rise of algae with one data point may be rather insecure, and as we point out in the amended manuscript "this single data point does not yet allow assessing the temporal and geographical extent of the algal-ecosystems and may not yet reflect a persistent and global change in marine primary production".

In lines 322 etc., the authors write: "In contrast to previous studies 4,58, we here posit that the rise of algae occurred not as a direct consequence of enhanced nutrient availability. Similar to the requirement for bioavailable oxygen for respiration 59,60, we consider enhanced phosphorus availability as a prerequisite but not as the trigger for the rise of algae. Instead, our data indicate that enhanced feeding pressure was needed to decimate the dominant cyanobacterial population¹⁰."

In the above quotation, the authors claim that reference 58 states that the rise of algae is a direct consequence of increasing nutrients. However, reference [58] in fact states the opposite:

"In the immediate aftermath of the melting glaciers, rising nutrient levels hypothetically resulted in a massive increase in cyanobacterial phytoplankton numbers until bacterivorous grazers, such as ciliates, radiated into the pelagic realm, capping bacterial biomass [22]. As in the modern ocean, this bacterivorous activity would have made nutrients available for larger phytoplankton [71] — and it is this effect that we may recognize as surging algal steranes in the Cryogenian."

This is in fact exactly what the authors claim as their new and contrasting explanation. The authors should thus rather first cite the original idea of [58] that a proliferation of ciliates (or other bacterivorous grazers) may have been the trigger for algal proliferation and THEN use their new data to show that the prediction in [58] may in fact have been correct. (see also earlier comment on this)

Thank you for reminding us of this. We did not attempt to hijack the concept in any way and apologize if it seemed this way. We have now amended this aspect and have correctly cited the initial idea.

The dolomite story

The new version is much less speculative and thus much better, but there is still one statement that appears implausible to me and that can simply be removed without damaging the paper: you may simply want to state that bacterial heterotrophy in some form associated with BNG formation, such as specific sulphate reducers, may be associated with dolomitization. 'Enhanced heterotrophy' is a weak explanation for reasons criticized by both reviewers in the first round of reviews: the heterotrophy in these sediments is by no means 'enhanced' relative to many organic lean limestones. Heterotrophy is just very unusual in these samples, not more complete in magnitude. Moreover, it is totally unclear in which way ciliates would or could have been involved in dolomitization. Do you know of any other occurrence of dolomite where BNG or gammacerane are elevated? Shouldn't most dolomites contain BNG then?

Indeed, we have clarified this statement and toned down the relevant section. We do not suspect an involvement of ciliates in mediating dolomite precipitation. The idea was rather that the extent of the degradation process could be relevant—for example acidic functionalities in DOM that forms as a side product during degradation. Given that DOM will form during heterotrophic reworking of primary produced organic matter (Javor, 1989) and such DOM will never end up in the sedimentary record, independent of its concentration, the 'almost complete' degradation of a large flux of primary produced organic matter will have different environmental consequences than the 'almost complete' degradation of a small flux of primary produced organic matter due to the scaling of remnant degradation products that may play a role.

365: Here the authors state that the shift from bacteria to eukaryotes occurred between the Sturtian and Marinoan. I agree with this, but it contradicts the earlier claim that it occurred after melting of the Marinoan Snowball.

Thank you for pointing this out, we have amended statements to make them internally consistent.

367: "eukaryotic steranes remain low during the Snowball Earth interglacial 4"

I think it would be wrong to cite [4] for this statement. [4] claims that steranes increased in the 'interglacial'. Moreover, the word 'interglacial' is reserved for short warm phases within glaciations, and this terminology is probably not appropriate for a 15 million year hot phase.

We have corrected the reference. Given that previous studies have referred to this period as the Cryogenian interglacial (e.g. Hayes et al., 1999; Ader et al., 2009; Halverson et al., 2010; Love and Summons, 2015) we deem it apt to continue this terminology. If the reviewer is aware of a more apt terminology for the period between the Sturtian and the Marinoan glaciation, we would be glad to follow it.

377-378 (and other places): While I agree with reviewer 3 that the Close and Pearson interpretation for the isotope offset needs to be cited and discussed, the Chuar data may rather speak in favour of the Logan model: in the Chuar, there does not appear to be any evidence for increased algal primary productivity in parallel with the isotope shift (or is there? If yes, then this data needs to be presented). If there is in fact no evidence for a shift from bacterial to eukaryotic primary productivity in parallel with the isotope shift, then this speaks against the Close and Pearson model (in this particular instance). By contrast, you do have evidence for heterotrophic reworking (25-nor hopanes) in parallel with the isotope shift, and this is actually quite a beautiful confirmation of the Logan model (at least in this data set). I think you should make a call for one or the other model when discussing the isotope data because there is no evidence that both mechanisms are at play reinforcing each other (as claimed in the SI text) – and it is simply not terribly likely.

To summarize the uncertainties in the current interpretation:

(1) The Chuar section shows a correlation between BNG and isotopic offset. There is evidence that degradation is responsible for this offset (Logan model) but no evidence for an increase in algal productivity (Close model).

(2) For the Araras section, it was not possible to measure isotopes, but you speculate that there may be the same correlation. Based on such a hypothetical correlation, you speculate that (a) heterotrophic reworking caused the BNG abundance (ala Logan) AND (B) that the hypothetical isotope shift would also have been caused by an increase in algal productivity (Close model) without sterane data to back this up.

There is no increase in sterane abundance and diversity in the Araras, and there is no data for an isotopic offset. Thus, the evidence is clearly too weak and speculative to conclude that there was an increase in algal productivity relative to bacterial primary productivity in the analyzed Araras section.

Due to the fact that the manuscript changed its focus throughout the review cycles, some of this data was not previously presented. We do in fact observe a connection between BNG, the relative abundance of steranes and degradation in the Araras Group. Then we have a connection between BNG, Dd13C and the relative abundance of steranes in the Chuar Group. We have altered both figure 1 (Araras) and figure 4 (Chuar) with improved plots to show how steranes become significantly more abundant in regions with little to no BNG. As we believe that both interpretations of Dd13C variation are correct: enhanced degradation and community shifts are not exclusive. In fact, as we now discuss throughout the manuscript, we consider all linked.

378: 'Enhanced heterotrophic reworking' is back. I would really remove it, it is just not true. Just cite highly unusual heterotrophic conditions.

Corrected

380: See above, this is not a new idea. Rather state that you confirm the hypothesis by ref [58].

Corrected

382: the idea of a return of cyanobacteria through elevated temperatures was postulated by ref [4], not [15]

Corrected

384: Here you acknowledge other papers for the feeding pressure idea. Good.

Thank you.

423: remove 'intense heterotrophic reworking' and maybe state 'suggested to indicate ciliate activity'

Corrected

424 after the MARINOAN Snowball Earth

Corrected

Figure 4: this data is really incredibly cool!

Thank you, we have further improved the figure by adding in the available sterane data.

Supplementary Material

125: world dominated by cyanobacteria: please check whether ref 23 really makes this statement.

Corrected

143: ref error

Corrected

144: same as in main text.

Corrected

149: sizes

Corrected

225: delete 'be'

Corrected

In summary, there are a number of important points that need to be addressed, but overall this is a superb, important and well written paper that is a major contribution to the field. Based on the beautifully executed biomarker identification and importance of the unusual biomarker distribution, it will become a classic.

Thank you, we are very grateful for your insightful comments, which have helped us to significantly improve the manuscript.

Reviewer #3 (Remarks to the Author):

The authors have done a considerable amount of re-writing to address the concerns of

this reviewer (and others). Most importantly, they have exorcised the problematic "extreme heterotrophy" references and included a discussion of the possibility of a direct precursor for BNG. I think they have struck a very nice balance in this regard. To be clear, there will probably still be significant debate in the community about their environmental interpretation of the BNG record. That is par for this course, and exactly what makes the manuscript so interesting. I think you should publish the manuscript as is. Its going to be a classic.

Thank you for your kind words. We highly appreciate the comments you provided during the first round of reviews and which have helped us to get the manuscript into a much better shape.

References:

- Ader, M., Macouin, M., Trindade, R.I.F., Hadrien, M., Yang, Z., Sun, Z., Besse, J., 2009. A multilayered water column in the Ediacaran Yangtze platform? Insights from carbonate and organic matter paired $\delta^{13}\text{C}$. *Earth and Planetary Science Letters* 288, 213–227.
- Brocks, J.J., Jarrett, A.J.M., Sirantoine, E., Hallmann, C., Hoshino, Y., Liyanage, T., 2017. The rise of algae in Cryogenian oceans and the emergence of animals. *Nature* 548, 578–581.
- Butterfield, N.J., 2000. *Bangiomorpha pubescens* n. gen., n. sp.: implications for the evolution of sex, multicellularity, and the Mesoproterozoic / Neoproterozoic radiation of eukaryotes. *Paleobiology* 26, 386–404.
- Close, H.G., Bovee, R., Pearson, A., 2011. Inverse carbon isotope patterns of lipids and kerogen record heterogeneous primary biomass. *Geobiology* 9, 250–265.
- Gibson, T.M., Shih, P.M., Cumming, V.M., Fischer, W.W., Crockford, P.W., Hodgskiss, M.S.W., Wörndle, S., Creaser, R.A., Rainbird, R.H., Skulski, T.M., Halverson, G.P., 2017. Precise age of *Bangiomorpha pubescens* dates the origin of eukaryotic photosynthesis. *Geology* 46, 135–138.
- Halverson, G.P., Wade, B.P., Hurtgen, M.T., Barovich, K.M., 2010. Neoproterozoic chemostratigraphy. *Precambrian Research* 182, 337–350.
- Hayes, J.M., Strauss, H., Kaufman, A.J., 1999. The abundance of in marine organic matter and isotopic fractionation in the global biogeochemical cycle of carbon during the past 800 Ma. *Chemical Geology* 161, 103–125.
- Javor, B., 1989. *Hypersaline Environments: Microbiology and Biogeochemistry*. Brock/Springer Series in Contemporary Bioscience.
- Love, G.D., Grosjean, E., Stalvies, C., Fike, D.A., Grotzinger, J.P., Bradley, A.S., Kelly, A.E., Bhatia, M., Meredith, W., Snape, C.E., Bowring, S.A., Condon, D.J., Summons, R.E., 2009. Fossil steroids record the appearance of Demospongiae during the Cryogenian period. *Nature* 457, 718–721.
- Love, G.D., Summons, R.E., 2015. The molecular record of Cryogenian sponges - A response to Antcliffe (2013). *palaeontology* 58, 1131–1136.

REVIEWERS' COMMENTS:

Reviewer #1 (Remarks to the Author):

The authors have replied very well to the last round of comment and I am more than happy with the changes. The addition of the sterane/hopane data is brilliant, absolutely brilliant – this really rounds the story up and makes it utterly convincing. Well done, beautiful work. This can now be published as is.